# SMaRT lncRNA controls translation of a G-quadruplex-containing mRNA antagonizing the DHX36 helicase

Julie Martone[1,†], Davide Mariani[2,†], Tiziana Santini[2], Adriano Setti[1], Sama Shamloo[1], Alessio Colantoni[1], Francesca Capparelli[1], Alessandro Paiardini[3], Dacia Dimartino[1], Mariangela Morlando[4] & Irene Bozzoni[1,2,*] (iD)

## Abstract

Guanine-quadruplexes (G4) included in RNA molecules exert several functions in controlling gene expression at post-transcriptional level; however, the molecular mechanisms of G4-mediated regulation are still poorly understood. Here, we describe a regulatory circuitry operating in the early phases of murine muscle differentiation in which a long non-coding RNA (SMaRT) base pairs with a G4-containing mRNA (Mlx-γ) and represses its translation by counteracting the activity of the DHX36 RNA helicase. The time-restricted, specific effect of lnc-SMaRT on the translation of Mlx-γ isoform modulates the general subcellular localization of total MLX proteins, impacting on their transcriptional output and promoting proper myogenesis and mature myotube formation. Therefore, the circuitry made of lnc-SMaRT, Mlx-γ, and DHX36 not only plays an important role in the control of myogenesis but also unravels a molecular mechanism where G4 structures and G4 unwinding activities are regulated in living cells.

**Keywords** G-quadruplex; lncRNAs; Myogenesis; RNA helicases; RNA–RNA interactions
**Subject Categories** Development; RNA Biology

## Introduction

Long non-coding RNAs (lncRNAs) belong to a complex class of transcripts which are expressed in all cell types and can be considered as key regulators of development and differentiation, thanks to the exquisite regulation of their spatiotemporal pattern of expression [1,2]. Loss- and gain-of-function experiments indicated that they may play an important role in normal development and differentiation as well as in many pathological conditions [3,4]; nevertheless, the identification of the precise mechanism of action is still lacking for most of them. The propensity to fold into complex secondary structures, together with their modular architecture that combines nucleic acid- and protein-binding domains into the same molecule, confers them the ability to perform a plethora of diverse functions [5]. The subcellular distribution of lncRNAs is an important determinant in understanding their functional role [6]: While nuclear species have been mainly linked to epigenetic modifications and transcriptional control, cytoplasmic lncRNAs take on a central role in different steps of post-transcriptional regulation of gene expression [7–9]. They were shown to act as decoy molecules for microRNAs or protein partners [10–12] and to mainly affect mRNA stability [13–15] and translation [16–19]. Although in some cases specific motifs of RNA–RNA pairing have been indicated at the basis of the lncRNA-target recognition [13,16,17], very little is known regarding the factors that contribute to the specificity and stabilization of these interactions. An important component in translational regulation is represented by G-quadruplex regions, non-canonical secondary structures that form within G-rich DNA or RNA sequences by Hoogsteen hydrogen bonds [20], which have been shown to require specific helicases to be solved [21].

Here, we show a regulatory circuitry controlled by a muscle-specific cytoplasmic lncRNA (lnc-SMaRT, Skeletal Muscle Regulator of Translation) which is essential for proper differentiation of murine myogenic precursors: By direct base pairing with a G-quadruplex region present in the Mlx-γ mRNA, it prevents its translation in an antagonistic manner with the RNA helicase DHX36. Notably, repression of the MLX-γ protein was found to control the nuclear localization of the other two MLX isoforms (α and β) and in turn regulate the expression of their target genes.

1  Department of Biology and Biotechnology, Charles Darwin, Sapienza University of Rome, Rome, Italy
2  Center for Life Nano Science@Sapienza, Istituto Italiano di Tecnologia, Rome, Italy
3  Department of Biochemical Sciences, Sapienza University of Rome, Rome, Italy
4  Department of Pharmaceutical Sciences, University of Perugia, Perugia, Italy
   *Corresponding author. Tel: +39 06 4991 2202; E-mail: irene.bozzoni@uniroma1.it
   †These authors contributed equally to this work as first authors

# Results

## Lnc-SMaRT depletion affects myoblast differentiation

Lnc-SMaRT (Skeletal Muscle Regulator of Translation) is an intergenic long non-coding RNA, previously named lnc-049 (MGI Symbol: GM14635), identified as a murine skeletal muscle species [22]. Lnc-SMaRT is composed of four exons (1409 nt, Fig 1A) and has a predominant cytoplasmic localization (Fig 1B). Its expression starts at day 1 of murine C2C12 muscle cell

differentiation, peaks at day 2, and decreases afterward (Fig 1C), mirroring the profile of the early myogenic markers MyoD and myogenin (Fig EV1A). The *in vivo* expression of lncSMaRT was analyzed in different tissues obtained from control mice and dystrophic *mdx* mutants, which are characterized by high levels of muscle regeneration [23]. The *mdx* condition was selected due to the observed *in vitro* involvement of lncSMaRT in early steps of myogenesis. PCR analyses showed that the expression of lncSMaRT occurs in *mdx* muscles while it is absent in skeletal and cardiac muscles of wild-type mice, again suggesting that

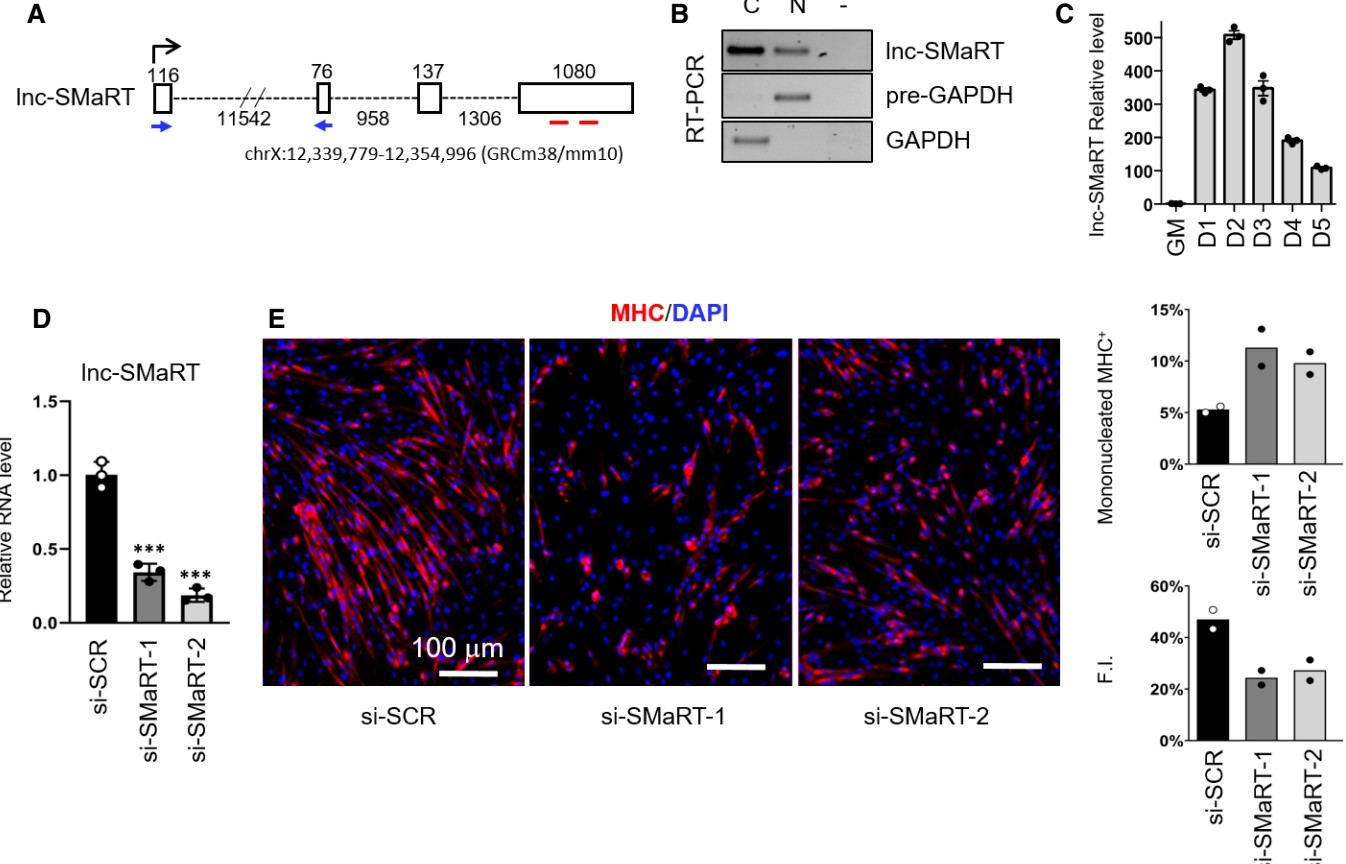

**Figure 1.  Lnc-SMaRT functional characterization.**

A   Schematic representation of the lnc-SMaRT genomic locus. Genomic coordinates and exon/intron lengths are indicated. qPCR primer location is indicated by blue arrows, while the regions corresponding to the siRNAs target sites are indicated by red lines.

B   RT–PCR on cytoplasmic (C) and nuclear (N) extracts showing the subcellular localization of lnc-SMaRT. RNA was isolated from C2C12 cells after 2 days of differentiation. GAPDH mRNA and pre-mRNA (pre-GAPDH) were used, respectively, as cytoplasmic and nuclear controls. Representative results from three independent experiments are shown.

C   qPCR analysis of lnc-SMaRT RNA expression in C2C12 cells undergoing differentiation at the indicated time points. The RNA expression levels in qPCR analyses were normalized against GAPDH mRNA and expressed as relative quantities with respect to GM samples set to a value of 1. Data are presented as the mean ± s.e.m. of three biological replicates (dots).

D   qPCR on RNA extracts from C2C12 myoblasts treated with either control siRNAs (si-SCR) or with two different siRNAs against lnc-SMaRT (si-SMaRT-1, si-SMaRT-2) and induced to differentiate for 2 days. The RNA expression levels in qPCR analyses were normalized against GAPDH mRNA and expressed as relative quantities with respect to GM samples set to a value of 1. Data are presented as the mean ± s.e.m. of three biological replicates (dots). Statistical analysis was performed with ordinary analysis of variance (ANOVA) followed by Dunnett's multiple comparison test. ***$P < 0.001$.

E   Representative immunofluorescence for myosin heavy chain (MHC in red) in combination with DAPI staining (in blue) C2C12 murine myoblasts transfected with either control (si-SCR) or lnc-SMaRT siRNAs (si-SMaRT-1, si-SMaRT-2) fixed after at 2 days of differentiation. Histograms represent MHC-positive mononucleated cells/total MHC-positive cells ratio and fusion index quantification (F.I.). At least 5 randomly chosen microscope fields of two independent biological samples were analyzed ($n > 600$ cells for each field). Data are presented as the mean of the two biological replicates (dots).

Source data are available online for this figure.

lncSMaRT plays a role also in *in vivo* muscle regeneration (Fig EV1B). Interestingly, 80% downregulation of lnc-SMaRT with two independent siRNAs (#1 and #2) in murine C2C12 cells (Figs 1D and EV1C) produced alteration of the myogenic process with a 50% increase of myosin heavy-chain mononucleated-positive cells ($MHC^+$) together with a 50% reduction of the fusion index (F.I., Fig 1E). Quantitative PCR (qPCR) analysis indicated that while the levels of the early myogenic factors MyoD and Myog were not affected, those of late differentiation markers such as myocyte enhancer factor 2C (Mef2C), myosin creatine kinase (Mck), and dystrophin (Dys) were downregulated (Fig EV1D and E). These data allowed the placement of lnc-SMaRT activity at an intermediate stage between early and late differentiation; indeed, lnc-SMaRT could be placed downstream of MyoD and Myog since these proteins were shown to bind the lnc-SMaRT putative promoter region (Chip seq data sets from Mouse ENCODE Project (http://www.ncbi.nlm.nih.gov/geo/query/acc.cgi?acc = GSE36024) inspected in [22]) and also because they were not affected in conditions of lnc-SMaRT depletion (Fig EV1D and E).

Total RNA-Seq analysis was performed on two independent populations of C2C12 cells treated with siRNA#1 and control scramble siRNA. This analysis showed that lnc-SMaRT (RPKM 17.46) is a medium-high abundant lncRNA being expressed only 3.6 times less than MyoD (RPKM 62.42) and 4 times less that lincMD1 (RPKM of 69.9), that is considered a fairly abundant muscle-specific lncRNA [24]. As shown in Fig EV2A, even if gene expression profiles of lnc-SMaRT-depleted samples were more dissimilar with respect to control samples, several genes were found to be consistently altered upon lnc-SMaRT depletion, with the majority of them (112 out of 184) being downregulated (Fig 2A and B, Dataset EV1). Gene Ontology (GO) term enrichment analysis showed that many downregulated genes are involved in muscle contraction and steroid biosynthesis, while the upregulated genes were mostly related to cell proliferation (Fig 2C, Dataset EV2). Selected deregulated mRNAs were validated by qPCR in samples independently treated with the two lnc-SMaRT siRNAs (#1 and #2). In particular, coherently with the RNA sequencing data and the observed phenotype, Lrrn1, Tnnc2, and Crabp2 mRNAs, physiologically upregulated during C2C12 differentiation [25,26], resulted downregulated (Fig EV2B).

In conclusion, these data showed that lnc-SMaRT plays an important role in controlling the correct timing of myoblast differentiation with a clear effect on genes involved in intermediate stages of myogenesis.

In the attempt to perform a rescue experiment and since plasmid transfection of C2C12 cells is very inefficient, we raised a stable C2C12 cell line overexpressing mature lnc-SMaRT amplified from cDNA under the control of eIF1a promoter. Unfortunately, the overexpressing line displayed an apoptotic phenotype which hindered such type of analysis: In fact, myoblasts overexpressing the lncRNA were impeded to enter the myogenic program upon serum starvation by displaying only few elongated and oriented cells and decrease in mRNA and protein levels of MyoD and myogenin (Fig EV2D). Notably, at 48 h of differentiation, we observed an apoptotic phenotype, as indicated by increased activation of caspase 3 (Fig 2D) and TUNEL assay (Fig EV2E). The increase in apoptosis and consequent reduction of the number of cells maintained in the myogenic program can explain the decrease in MyoD and myogenin

as well as the increase of transcripts related to cell proliferation observed upon lnc-SMaRT depletion. Altogether, these data indicate that the amount and timing of lnc-SMaRT expression should be finely regulated to establish a correct myogenic program.

## Identification of lnc-SMaRT interactors

To identify the binding partners of lnc-SMaRT, RNA pull-down experiments were performed with two sets of biotin-labeled DNA antisense oligonucleotides (Set#1 and Set#2, Fig 3A) on extracts derived from C2C12 cells at day 2 of differentiation. A set of antisense oligonucleotides against LacZ mRNA (LacZ), with a similar GC content, was used as a negative control. Proteins and RNA purified after streptavidin capture of the molecular complexes were subjected to mass spectrometry (MS) and NGS analysis, respectively. The MS data allowed to short list three *bona fide* interactors (PURB, IQGA1, and DHX36) based on the number of unique peptides (more than 5) and on the enrichment with both sets of specific oligos in comparison with LacZ probes. PUR β is a single-stranded DNA- and RNA-binding protein that has been previously involved in DNA replication/transcription and in mRNA translation [27], while IQGA1 is a Ras GTPase-activating-like protein that belongs to a family of scaffolding proteins involved in several cellular processes such as cell cycle regulation, cell–cell adhesion, and actin cytoskeleton organization [28]. The ATP-dependent RNA helicase DHX36 was selected because of the absence of peptides in the LacZ sample (Dataset EV3). This enzyme had been previously shown to bind and unwind G-quadruplex (G4) structures in both DNA and RNA molecules [29–31]. Western blot analysis with DHX36 antibodies validated the strong and specific enrichment of this protein in the lnc-SMaRT pull-downs obtained with both sets of specific probes (Fig 3B). Finally, RNA immunoprecipitation (RIP) performed with DHX36 antibodies and appropriate controls further confirmed the association of the helicase with lnc-SMaRT (Fig 3C).

We also performed RNA-seq analysis of the RNAs recovered from lnc-SMaRT pull-down; apart from lnc-SMaRT, which served as a positive control, we identified a set of significantly enriched transcripts deriving from 17 protein-coding genes (Table EV1). We did not find any GO terms enriched in this list; however, 12 out of 17 genes resulted to encode for mRNAs with a predicted G-quadruplex (QGRS Mapper software [32], Table 1). Since DHX36, a protein known to solve RNA G4 structures, was found among the major interactors of lnc-SMaRT, we concentrated on this class of transcripts. Among them, we selected the Mlx mRNA due to its known role in controlling myogenesis through the induction of several myokines [33]. Moreover, such a transcript is present in three isoforms (α, β, and γ) that have altogether an expression level (RPKM 23.88) similar to that of lnc-SMaRT (RPKM 17.46). Notably, all the three isoforms (α, β, and γ) resulted enriched in the pull-down of lnc-SMaRT (Fig 3D). Pull-down experiments performed in psoralen-crosslinking conditions allowed to further confirm the direct pairing between Mlx mRNAs and lnc-SMaRT (Fig EV3A, left panel). Notably, when the psoralen-crosslinking pull-down was performed upon the depletion of lnc-SMaRT, the enrichment of Mlx mRNA was lost (Fig EV3A, right panel).

Sequence analysis indicated that the Mlx mRNAs display a double interaction with lnc-SMaRT: all three isoforms contain an extended region of complementarity in their 3′UTR, while only the

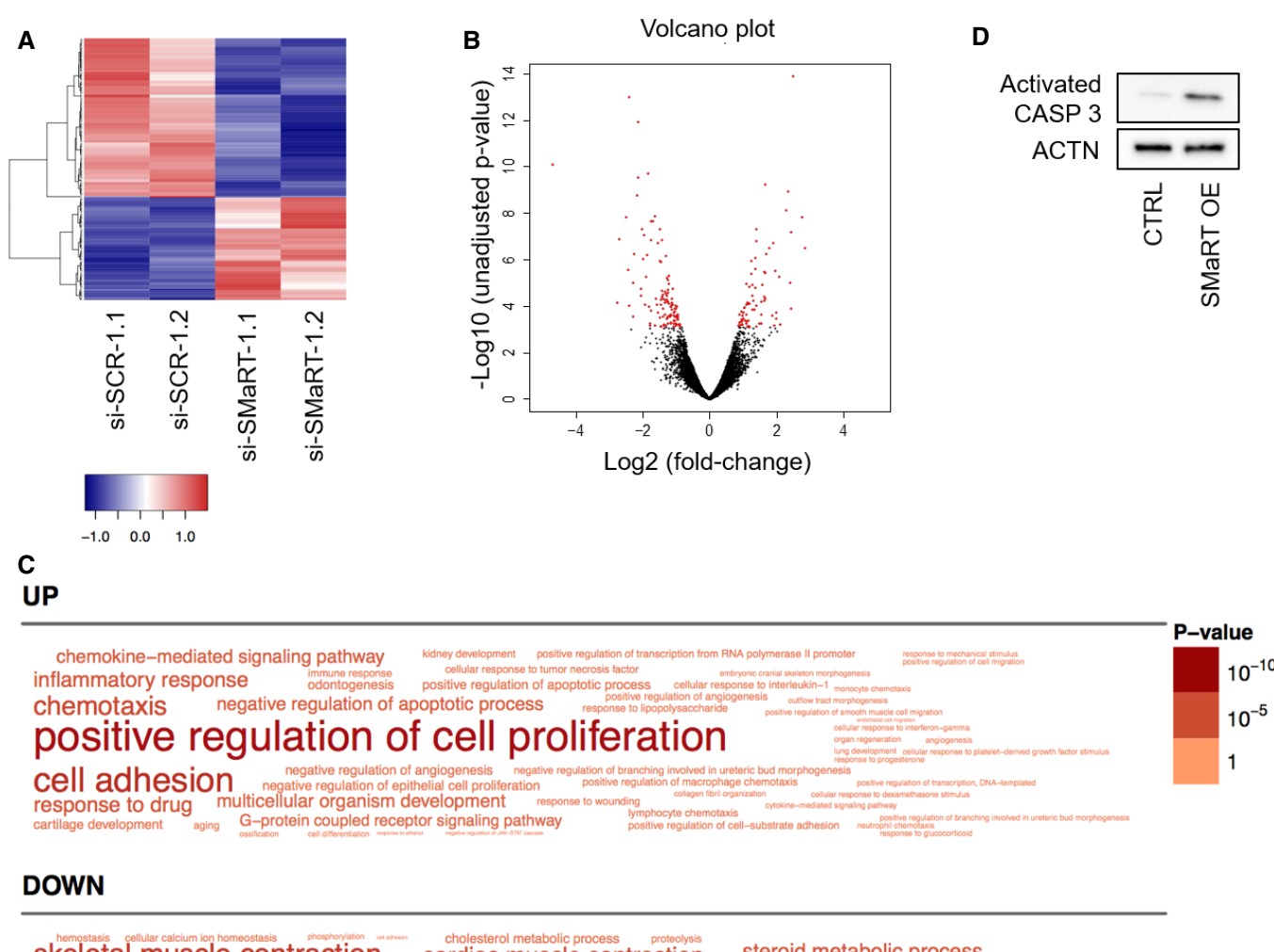

**Figure 2. lnc-SMaRT depletion impacts on C2C12 transcriptome.**

A   Gene expression heatmap showing the hierarchical clustering of protein-coding genes whose expression is altered upon lnc-SMaRT depletion. Expression values, calculated as RPKMs, were $\log_2$-transformed and mean-centered. Only genes with an average RPKM between 10 and 1,000 were plotted.

B   Volcano plot describing the differential gene expression between si-SCR- and si-SMaRT-treated C2C12 samples. For each gene, $\log_2$ of fold-change versus $-\log_{10}$ of the unadjusted $P$-value is plotted. Points in red represent differentially expressed genes (FDR-adjusted $P$-value < 0.05).

C   Word cloud representing the Gene Ontology Biological Process terms enriched (Benjamini–Hochberg-corrected $P$-value < 0.05) in the lists of genes upregulated (UP) or downregulated (DOWN) upon lnc-SMaRT knockdown. The size of the words correlates with the Benjamini–Hochberg-corrected $P$-value.

D   Western blot on protein extract from control and lnc-SMaRT overexpressing C2C12 stable cell lines collected after 1 day of differentiation. CASP3 antibody was used. ACTN was used as loading control. Representative results from three independent experiments are shown.

Source data are available online for this figure.

Mlx-γ isoform has a second pairing region located in its specific first exon (Fig EV3B and C, see pairing of Mlx isoforms with A and B regions of lnc-SMaRT). When analyzed with the QGRS Mapper software [32], this region resulted to contain a *bona fide* G4 element. Notably, immunoprecipitation with DHX36 antibody showed that only the Mlx-γ isoform was enriched in the IP fraction (Fig 3E, upper panel) and that this enrichment was maintained also in the absence of lnc-SMaRT (Fig 3E, lower panel). These results indicated that Mlx-γ is the only isoform able to interact with the helicase,

likely through the G4 element, and that this interaction can occur even in the absence of lnc-SMaRT. When RNAi against lnc-SMaRT or DHX36 was applied to C2C12 cells induced to differentiate, no differences in the levels of the three Mlx mRNA isoforms were found (Fig EV3D), prompting us to test whether any regulation could occur at the translational level. Since the three MLX isoforms are difficult to be distinguished on Western blots, we raised plasmid constructs expressing flagged versions of MLX-α, MLX-β, and MLX-γ and tested their behavior in conditions of lnc-SMaRT

overexpression. HeLa cells were selected since they can be efficiently transfected and because, while expressing DHX36, they are devoid of lnc-SMaRT. The overexpression of lnc-SMaRT produced a consistent decrease of Mlx-γ translation and not of the α- and β-isoforms (Fig 3F, upper and middle panels), without affecting the RNA levels (Fig 3F, lower panel), indicating that lnc-SMaRT acts as a repressor of Mlx-γ translation. Downregulation of DHX36 (Fig EV3E) also reduced the MLX-γ protein (Fig 3G, upper panel), while no effects were observed at the RNA level (Fig 3G, lower panel), indicating that the helicase is indeed required for promoting Mlx-γ translation. In conclusion, these data indicate that DHX36 and lnc-SMaRT act in an antagonistic manner on Mlx-γ translation.

### Lnc-SMaRT controls the subcellular localization of Mlx proteins

Due to the absence of specific antibodies for MLX-γ and since this isoform was previously shown to re-localize MLX-α and MLX-β into the nucleus upon homodimerization [34], we decided to check by

*in situ* immunofluorescence whether alterations in MLX-γ levels could in turn affect the localization of total MLX proteins. We performed such assay in C2C12 cells at day 2 of differentiation in conditions of Mlx-γ mRNA depletion (Fig EV4A). The results indicate that, with respect to scramble siRNAs (panel si-SCR), the downregulation of Mlx-γ (panel si-Mlx-γ) produced a clear decrease in the nuclear MLX mean fluorescence (Fig 4A). The same effect was obtained when DHX36 depletion was performed (panel si-DHX36), indicating that the helicase controls the MLX-γ protein levels. When the same analysis was performed upon lnc-SMaRT depletion (panel si-SMaRT), a significant increase of the nuclear MLX mean fluorescence was observed. The quantifications of the nuclear versus cytoplasmic MLX mean fluorescence ratios are indicated in Fig 4B. Notably, no alteration of the Mlx-γ mRNA levels was observed in conditions of lnc-SMaRT and DHX36 depletion (Fig EV4A). Finally, the double knock down of Mlx-γ and lnc-SMaRT showed that the nuclear/cytoplasmic ratio of the MLX proteins was rescued to control levels (Fig EV4B). Altogether, these data show that MLX-γ is required for the nuclear accumulation of total MLX proteins and that the same effect is obtained upon

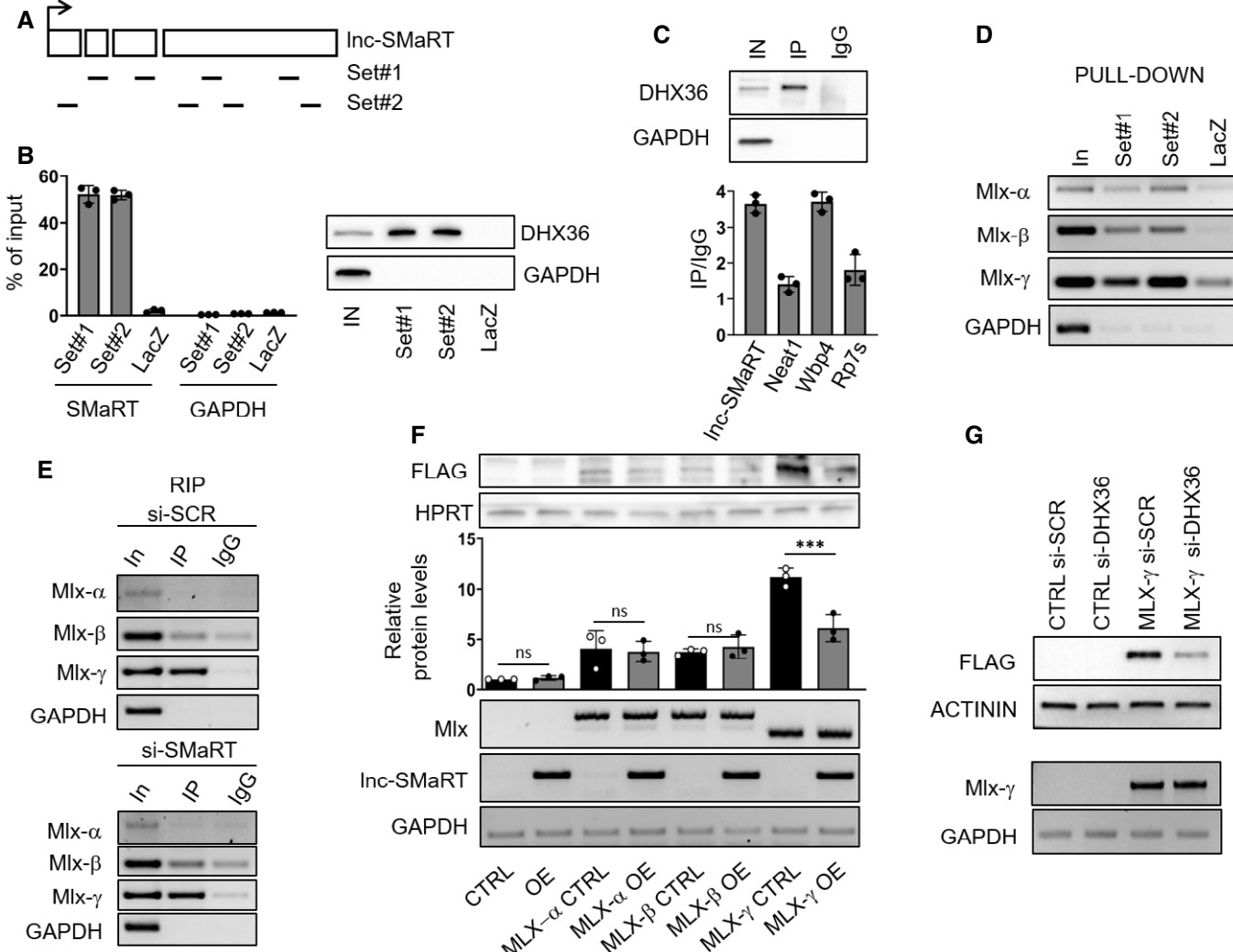

Figure 3.

**Figure 3. Lnc-SMaRT molecular interactome.**

A   Localization on the lnc-SMaRT transcript of the two sets of biotinylated probes (Set#1 and Set#2) used for lnc-SMaRT pull-down experiment.

B   Left panel: qPCR analysis of lnc-SMaRT enrichment in the RNA pull-down performed in C2C12 cells at day 2 of differentiation (D2); Set#1 and Set#2 probes were used against lnc-SMaRT together with a control set of probes against LacZ mRNA (LacZ). Data are expressed in percentage of input and presented as the mean ± s.e.m. of three biological replicates (dots). Right panel: Western blot analysis showing the specific enrichment of the DHX36 helicase in lnc-SMaRT pull-down; GAPDH was used as negative control. Representative results from three independent experiments are shown.

C   Upper panel: Western blot with DHX36 antibodies on protein extracts from DHX36 RNA immunoprecipitation. Input sample (IN) accounts for 2.5% of the extract. Representative results from three independent experiments are shown. Lower panel: qPCR analysis of lnc-SMaRT enrichment in DHX36 RIP-derived RNA extracts. WBP4 was used as positive control [31], while Neat1 (a lncRNA expressed at comparable level of lnc-SMaRT according to the RNA sequencing) and Rps7 were used as negative controls. Data are expressed as percentage of input normalized on IgG control and presented as the mean ± s.e.m. of three biological replicates (dots).

D   RT–PCR validation of Mlx-α, Mlx-β, and Mlx-γ mRNA enrichment upon the lnc-SMaRT pull-down performed with Set#1 and Set#2 probes; a control set of probes against LacZ mRNA (LacZ) was used as negative control. GAPDH was used as negative control. Input sample (IN) accounts for 10% of the extract. Representative results from three independent experiments are shown.

E   RT–PCR analysis of Mlx-α, Mlx-β, and Mlx-γ mRNA enrichment in DHX36 RIP-derived RNA extracts in samples treated with control siRNA (si-SCR, upper panel) or siRNA against lnc-SMaRT (si-SMaRT, lower panel). GAPDH was used as negative control. Input sample (IN) accounts for 10% of the extract. Representative results from three independent experiments are shown.

F   Upper panel: Western blot with FLAG antibody on protein extracts from HeLa cells overexpressing the indicated isoforms of FLAG-tagged MLX (MLX-α, MLX-β, and MLX-γ) in control condition (CTRL) or in overexpression of lncSMaRT (OE). FLAG tag has been inserted at the N-terminus of MLX protein isoforms. HPRT was used as loading control. Representative results from three independent experiments are shown. Middle panel: Quantification of FLAG-tagged protein levels normalized on HPRT signals. Data are expressed as the mean ± s.d. of three biological replicates (dots). Statistical analysis was performed with two-way ANOVA followed by Bonferroni's multiple comparisons test. ***$P < 0.001$. Lower panel: RT–PCR analysis of lnc-SMaRT and Mlx expression on RNA extracts from HeLa cells overexpressing the indicated isoforms of flagged MLX (MLX-α, MLX-β, and MLX-γ) in control condition (CTRL) or in overexpression of lncSMaRT (OE). GAPDH was used as loading control. Representative results from three independent experiments are shown.

G   Upper panel: Western blot with FLAG antibodies on protein extracts from N2a cells transfected with an empty vector (CTRL) or with a plasmid overexpressing FLAG-tagged MLX-γ (MLX-γ) in control conditions (si-SCR) or upon knockdown of DHX36 (si-DHX36). ACTININ was used as loading control. Representative results from three independent experiments are shown. Lower panel: RT–PCR analysis of Mlx-γ expression on the described RNA extracts. GAPDH was used as loading control. Representative results from three independent experiments are shown.

Source data are available online for this figure.

downregulation of lnc-SMaRT. Therefore, these results pointed to an inverse correlation between lnc-SMaRT levels and those of the MLX-γ protein. Further evidence for this anti-correlation is derived from immunofluorescence experiments where the nuclear localization of MLX was studied along C2C12 differentiation. The results indicated a clear inverse correlation between the amount of nuclear MLX protein and lnc-SMaRT expression; in fact, the lowest levels of nuclear MLX were observed at the time point when lnc-SMaRT expression was maximum (Fig EV4C and D). This occurred despite the unaltered levels of Mlx-γ mRNA (Fig EV4E).

Finally, the link between nuclear localization of MLX and its transcriptional activity was tested by intersecting its target genes in C2C12 cells (identified in [33]) with those deregulated upon lnc-SMaRT depletion. We found that the expression of a significant number of genes changed in the opposite direction upon the depletion of MLX and lnc-SMaRT (Fig 4C). qPCR analysis of CCL2 and CCL7, which are upregulated by MLX, showed their upregulation upon lnc-SMaRT depletion at day 2 of differentiation (Fig EV4F and Dataset EV1). In conclusion, these results suggest that lnc-SMaRT could act as a repressor of Mlx-γ translation and that in turn this directly affects the nuclear localization of total MLX proteins and their transcriptional output.

**Lnc-SMaRT controls the translation of Mlx-γ through direct base pairing with the G4-containing region**

To assess the involvement of lnc-SMaRT in Mlx regulation and the role of the two distinct interacting regions, luciferase assays were performed using three different reporter constructs (Fig 5A, left panel). RLuc-Mlx 5′ harbors the G4-containing region present in the first exon of Mlx-γ, cloned in-frame to the Renilla luciferase ORF (RLuc), mimicking its position in the corresponding mRNA.

RLuc-Mlx 3′ contains the 3′UTR of Mlx (that includes the second region of interaction with lnc-SMaRT), cloned downstream of the RLuc ORF, while RLuc-Mlx 5′-3′ carries both elements. Each reporter construct was co-transfected in proliferating C2C12 cells (where lnc-SMaRT is not expressed) with either an empty vector or with a plasmid expressing lnc-SMaRT (Fig EV5A, right panel). Figure 5A (right panel) shows that, with respect to the activity measured in the absence of lnc-SMaRT (#1), a 27% decrease of luciferase levels was reproducibly obtained when RLuc-Mlx 5′-3′ was co-transfected with lnc-SMaRT (#2). Such effect was reproduced with RLuc-Mlx 5′ (#3) and not with RLuc-Mlx 3′ (#4). Notably, in all cases the luciferase mRNA levels were not altered (Fig EV5A, left panel). These data indicated that lnc-SMaRT is able to mediate specific translational repression on the luciferase constructs and that the region containing the G4 element is the mediator of such regulation.

To verify the contribution of DHX36 in the regulation of RLuc-Mlx 5′, the luciferase assay was performed in cells treated with siRNAs against DHX36, with or without the overexpression of lnc-SMaRT (Fig EV5B). As shown in Fig 5B, the luciferase expression was reduced when samples were depleted of DHX36, independently from the presence of lnc-SMaRT. These data indicate that DHX36 plays a positive role in the translational control of the G4 element. Interestingly, in conditions of DHX36 depletion, the presence of lnc-SMaRT was able to further decrease luciferase activity, indicating that base pairing *per se* plays a negative role on translation.

To further investigate the nature of the interactions leading to translational repression, several other luciferase constructs were made. A derivative of RLuc-Mlx 5′ was produced by deleting 75 nucleotides encompassing the whole interacting region (RLuc-MlxΔ75, Fig 5C), including the G4-element. When this latter construct

was compared to the behavior of RLuc-Mlx 5′ in the absence (#1) or in the presence of lnc-SMaRT (#2), we observed the loss of responsiveness to lnc-SMaRT repression (#3) with the recovery of luciferase activity up to control levels. An additional construct was obtained by replacing 30 nucleotides of the Mlx-γ sequence containing the G4-element, with the corresponding pairing region of lnc-SMaRT (Ruc-Mlx$_{mut}$30). This construct failed to respond to lnc-SMaRT repression (#4); however, when the base pairing was restored by inserting into

**Table 1. G-quadruplexes predicted on transcripts from protein-coding genes recovered from lnc-SMaRT pull-down.**

| Gene | ID | Start | End | G4 | G-score | Localization |
|------|-----|-------|-----|----|---------|--------------|
| Acad8 | ENSMUST00000060513 | 299 | 334 | GGGATTTGGGGGGGTCTATGTGCGAACAGATGTGGG | 52 | CDS |
| Acad8 | ENSMUST00000120367 | 1 | 15 | GGGGGGCGGGGCGGG | 62 | 5′UTR |
| Acad8 | ENSMUST00000120367 | 316 | 351 | GGGATTTGGGGGGGTCTATGTGCGAACAGATGTGGG | 52 | CDS |
| Acsl6 | ENSMUST00000108905 | 24 | 48 | GGGGCTGCGGGGCTGCGGGCCTGGG | 62 | 5′UTR |
| Acsl6 | ENSMUST00000127731 | 2,807 | 2,838 | GGGTTGGGATTCTGGGTGTTCTCCATGGAGGG | 56 | / |
| Acsl6 | ENSMUST00000127731 | 2,615 | 2,654 | GGGTGGGATGGGGTAGTTCATGTCTAGGGTTGAGAGTGGG | 60 | / |
| Acsl6 | ENSMUST00000108904 | 24 | 48 | GGGGCTGCGGGGCTGCGGGCCTGGG | 62 | 5′UTR |
| Arf5 | ENSMUST00000020717 | 1,013 | 1,056 | GGGGGTACCCTTGGGGCCAGGTTTTGGGGGGAGGAAAGTGAGGG | 63 | 3′UTR |
| Coq2 | ENSMUST00000126981 | 20 | 51 | GGGAGGCGCGGGGCTCGCGCGGGGCCTGCGGG | 63 | / |
| Coq2 | ENSMUST00000126981 | 114 | 155 | GGGGTTCCGGGCGCGCGGGATCGGCGAGCCCCGGCCCCCGGG | 54 | / |
| Coq2 | ENSMUST00000135146 | 21 | 62 | GGGGTTCCGGGCGCGCGGGATCGGCGAGCCCCGGCCCCCGGG | 54 | / |
| Coq2 | ENSMUST00000031262 | 55 | 86 | GGGAGGCGCGGGGCTCGCGCGGGGCCTGCGGG | 63 | CDS |
| Coq2 | ENSMUST00000031262 | 149 | 190 | GGGGTTCCGGGCGCGCGGGATCGGCGAGCCCCGGCCCCCGGG | 54 | CDS |
| Glis3 | ENSMUST00000065113 | 2,048 | 2,086 | GGGCACTCCCCAGGGCCGGGGCCTGGGCCAGGGCCTGGG | 64 | / |
| Glis3 | ENSMUST00000162022 | 2,667 | 2,705 | GGGCACTCCCCAGGGCCGGGGCCTGGGCCAGGGCCTGGG | 64 | CDS |
| Glis3 | ENSMUST00000162022 | 7,280 | 7,322 | GGGGATGGTGATTATAATTAAAAGCAGATGGGGGGGGAAGGGG | 67 | 3′UTR |
| Glis3 | ENSMUST00000161026 | 1,713 | 1,751 | GGGCACTCCCCAGGGCCGGGGCCTGGGCCAGGGCCTGGG | 64 | / |
| Mlx | ENSMUST00000017945 | 98 | 114 | GGGGAGGGCGGGTCGGG | 63 | CDS |
| Myo1c | ENSMUST00000102505 | 4,451 | 4,485 | GGGTGCCTCTGTGACCTGGGAGCCTAGGGACAGGG | 56 | 3′UTR |
| Myo1c | ENSMUST00000108431 | 4,519 | 4,553 | GGGTGCCTCTGTGACCTGGGAGCCTAGGGACAGGG | 56 | 3′UTR |
| Myo1c | ENSMUST00000108431 | 4,402 | 4,422 | GGGAGCTACCGGGTGGGAGGG | 60 | 3′UTR |
| Myo1c | ENSMUST00000108431 | 280 | 324 | GGGCAGCGGAGCGGGGCGCCGGGTCCGGCAGGATGCGCTACCGGG | 54 | 5′UTR-CDS |
| Myo1c | ENSMUST00000108431 | 200 | 230 | GGGGCCTGCAAGGGGCGGTGCAGGGGGCGGG | 60 | 5′UTR |
| Myo1c | ENSMUST00000102504 | 4,458 | 4,492 | GGGTGCCTCTGTGACCTGGGAGCCTAGGGACAGGG | 56 | 3′UTR |
| Myo1c | ENSMUST00000069057 | 4,229 | 4,249 | GGGAGCTACCGGGTGGGAGGG | 60 | 3′UTR |
| Myo1c | ENSMUST00000069057 | 4,346 | 4,380 | GGGTGCCTCTGTGACCTGGGAGCCTAGGGACAGGG | 56 | 3′UTR |
| Myo1c | ENSMUST00000102505 | 4,334 | 4,354 | GGGAGCTACCGGGTGGGAGGG | 60 | 3′UTR |
| Myo1c | ENSMUST00000102504 | 4,341 | 4,361 | GGGAGCTACCGGGTGGGAGGG | 60 | 3′UTR |
| Ndrg4 | ENSMUST00000041318 | 2,377 | 2,413 | GGGCTGGAGATTGCCTGGCCCTTGGGTGGGAAATGGG | 51 | 3′UTR |
| Ndrg4 | ENSMUST00000080666 | 2,014 | 2,050 | GGGCTGGAGATTGCCTGGCCCTTGGGTGGGAAATGGG | 51 | 3′UTR |
| Ndrg4 | ENSMUST00000166358 | 2,243 | 2,279 | GGGCTGGAGATTGCCTGGCCCTTGGGTGGGAAATGGG | 51 | / |
| Ndrg4 | ENSMUST00000073139 | 2,079 | 2,115 | GGGCTGGAGATTGCCTGGCCCTTGGGTGGGAAATGGG | 51 | 3′UTR |
| Rbm45 | ENSMUST00000046389 | 41 | 75 | GGGGCGAGACGGGGAGCTGCCGGGAAGCGGCCGGG | 63 | 5′UTR |
| Six1 | ENSMUST00000050029 | 329 | 366 | GGGGCGGCAGGGTGGCGCGGCTTTGCTGCCGGGCCGGG | 53 | 5′UTR |
| Six1 | ENSMUST00000050029 | 1,599 | 1,635 | GGGTTCCTAAGTGGGGAGATATTGGGGCCTTGAAGGG | 63 | CDS-3′UTR |
| Spire1 | ENSMUST00000115050 | 253 | 297 | GGGCCCGGTTCTGGGTACAAGTGATGAGGGATTTGCGAAATGGGG | 62 | CDS |
| Spire1 | ENSMUST00000082243 | 407 | 451 | GGGCCCGGTTCTGGGTACAAGTGATGAGGGATTTGCGAAATGGGG | 62 | CDS |
| Spire1 | ENSMUST00000045105 | 351 | 395 | GGGCCCGGTTCTGGGTACAAGTGATGAGGGATTTGCGAAATGGGG | 62 | CDS |
| Usp10 | ENSMUST00000144458 | 2,260 | 2,295 | GGGCAAGGGCAGCGAGGACGAGTGGGAGCAAGTGGG | 56 | CDS |
| Usp10 | ENSMUST00000108988 | 1,803 | 1,838 | GGGCAAGGGCAGCGAGGACGAGTGGGAGCAAGTGGG | 56 | CDS |

lnc-SMaRT the complementary Mlx-γ sequence, the luciferase activity was strongly affected (#5). The luciferase mRNA levels, in all the analyzed conditions, were unchanged (Fig EV5C).

These data indicated that the complementarity between the A region of lnc-SMaRT and the G4-containing region of Mlx-γ is required to repress translation in a very specific manner.

**Region A of lnc-SMaRT is able to pair with G4 elements *in vitro***

Synthetic RNA oligonucleotides containing the G4 region of Mlx-γ (γ-oligo) and the complementary lnc-SMaRT sequence

(SMaRT-oligo) were used to test their *in vitro* ability to form G4 structures and to base pair (see schematic representation of Fig 6A). Their behavior was analyzed on polyacrylamide gels in both denaturing and native conditions. In denaturing conditions, using SYBR Gold staining, which allows the visualization of total RNA, a single band was observed for each oligo (Fig 6A, lanes #1 and #2). Instead, in native conditions, two different bands were detected with the γ-oligo: One migrated as the linear form (γ-oligo), while the other showed slower mobility, possibly due to a putative G-quadruplex structure (lane #4, γ-G4 band). This was confirmed by the use of a quadruplex-specific fluorescent

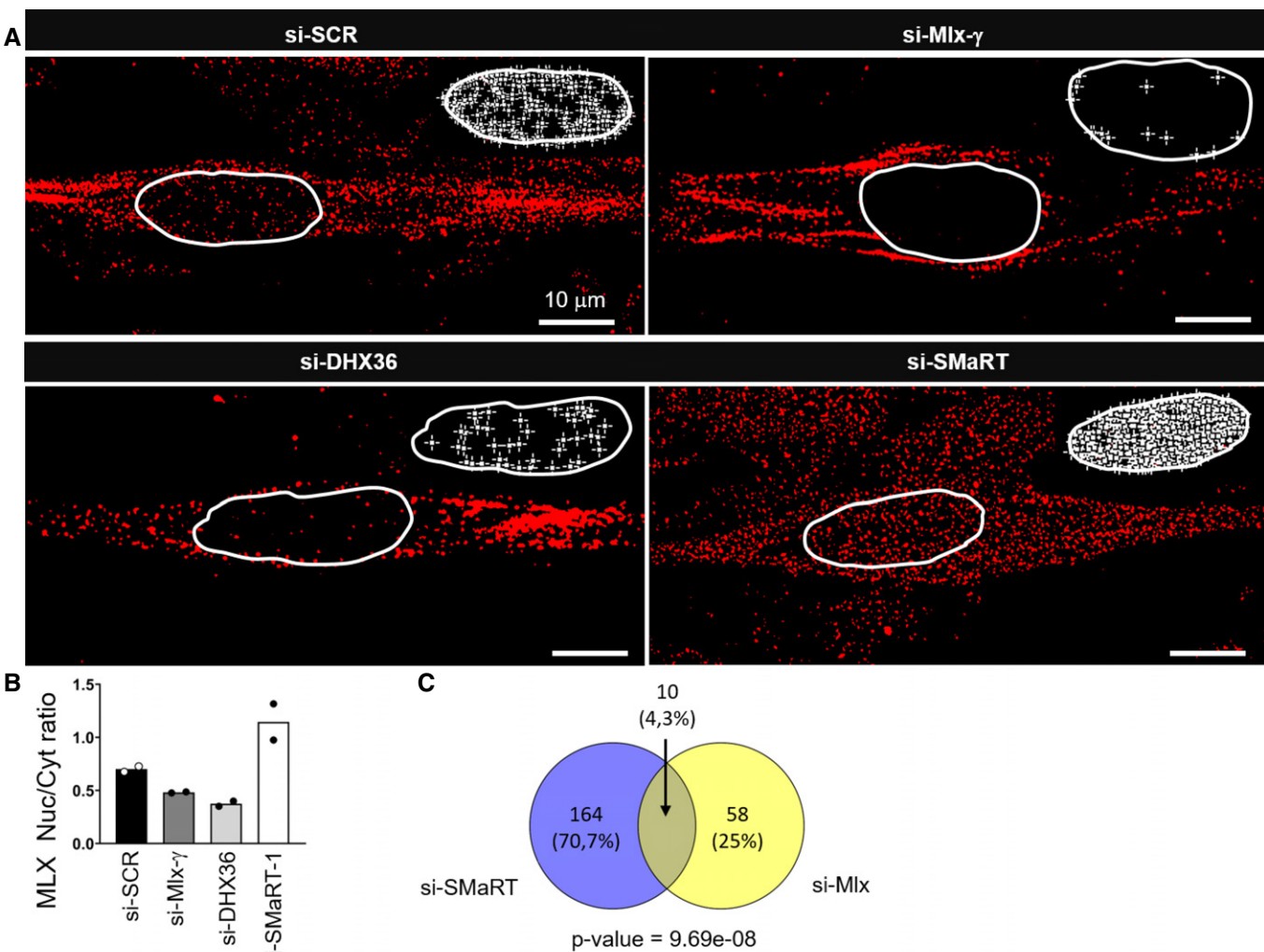

**Figure 4. Representative images for MLX immunofluorescence on C2C12 myotubes.**

A   Immunofluorescence for total MLX protein detection (red signal) performed on C2C12 cells treated with siRNAs against Mlx-γ (si-Mlx-γ), lnc-SMaRT (si-SMaRT), DHX36 (si-DHX36), or with a control siRNA (si-SCR). Cells were fixed 2 days after the switch to differentiation medium. Representative binary images of single confocal planes for MLX immunofluorescence are shown for each treatment. The inserts on the topside of the images sketch the MLX signal peaks in the nuclei (white stars) to highlight the variation of MLX staining in each condition. Dashed lines indicate the edge of the nucleus.

B   Histogram represents the fluorescence intensity ratio of the MLX protein signals in the nuclear (Nuc) and cytoplasmic (Cyt) compartment in the indicated conditions. About 50 cells from two independent experiments were analyzed as indicated in Materials and Methods. Data are presented as the mean of two biological replicates (dots).

C   Venn diagram showing the overlap between genes deregulated upon lnc-SMaRT depletion and MLX targets [33], both in C2C12 samples. Genes at the intersection are those with opposite fold-changes in the two depletion experiments. Only genes expressed in both systems were used to evaluate the overlap. Significance of overlap was evaluated using hypergeometric test.

dye (NMM, N-methyl mesoporphyrin IX [35,36]), which revealed specific staining only of the γ-G4 band (lane #6). Addition of KCl, known to stabilize G4 structures [37], increased the γ-G4 band upon NMM staining (Fig 6B, #2 vs #3), further supporting the presence of a G4 structure. Differently from the γ-oligo,

NMM staining did not reveal any putative G4 structure on the SMaRT-oligo (Fig 6A, lane #5).

When the γ- and SMaRT- oligos were mixed, denatured at 100°C, and analyzed on native gel after slow renaturation, several additional bands appeared (Fig 6A, lane #8). All of them showed lower

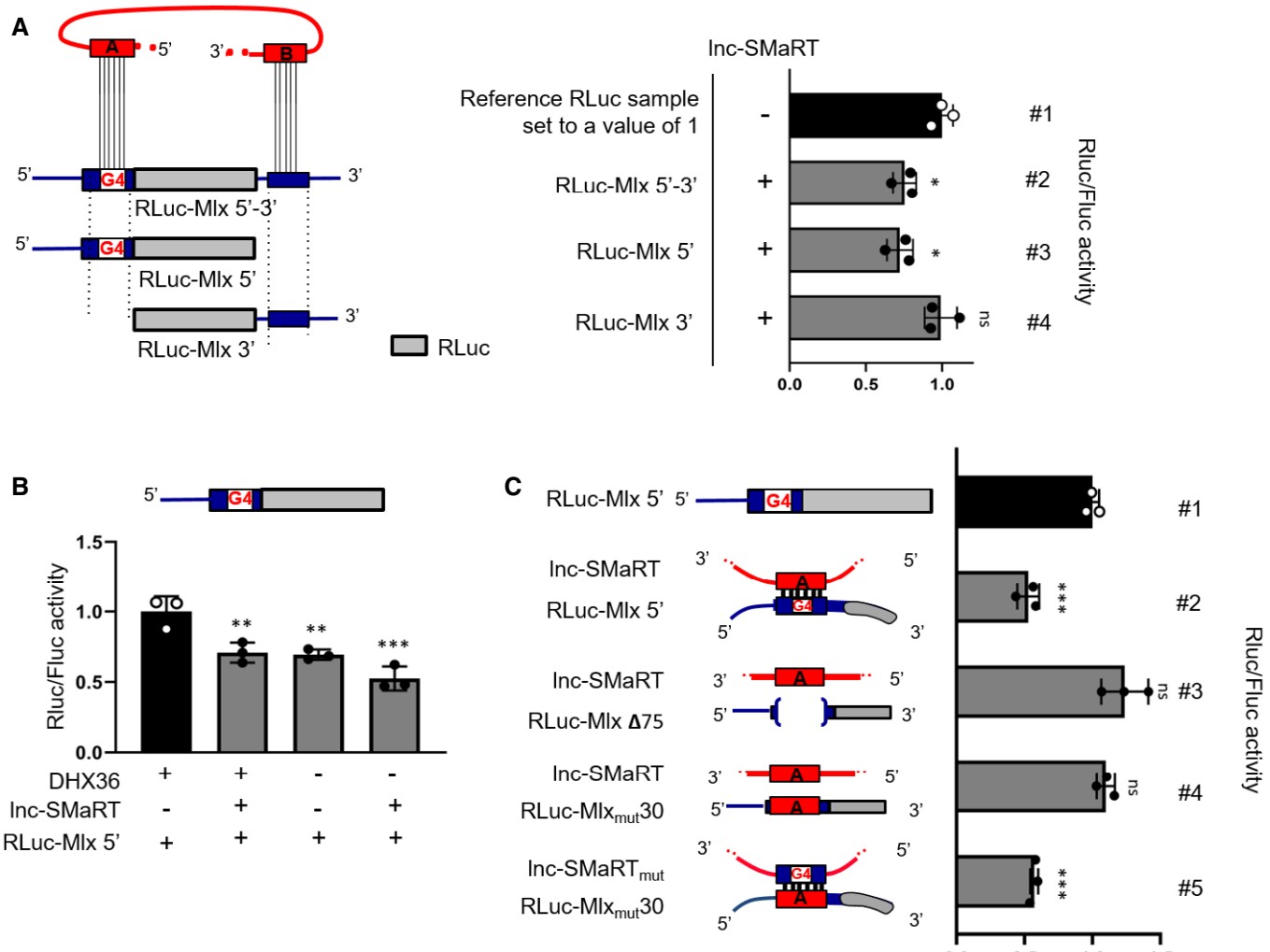

**Figure 5. Lnc-SMaRT controls the translation of Mlx-γ through direct base pairing with the G4-containing region.**

A  Left panel: schematic representation of the luciferase constructs produced for Mlx. Mlx 5′- and 3′UTR-interacting regions were cloned, respectively, upstream and downstream the Renilla luciferase coding region (RLuc-Mlx 5′-3′) and mutant derivatives devoid of the 5′ interacting region (RLuc-Mlx3′) or the 3′ UTR-interacting region (RLuc-Mlx5′) were obtained. These constructs were co-transfected in C2C12 myoblasts in growth conditions together with plasmids expressing lnc-SMaRT (+) or with a control vector (−). Right panel: Luciferase activity data, presented as the mean ± s.e.m. of three biological replicates (dots), are shown with respect to each RLuc control vector (RLuc-Mlx 5′–3′, RLuc-Mlx3′, RLuc-Mlx5′) set to a value of 1. Statistical analysis was performed with ordinary analysis of variance (ANOVA) followed by Dunnett's multiple comparison test. *$P < 0.05$.

B  The RLuc-Mlx5′ construct was co-transfected in N2a cells together with plasmids expressing lnc-SMaRT (+) or with a control vector (−) and treated with control siRNA (+) or siRNA against DHX36 (−). Luciferase activity data, presented as the mean ± s.e.m. of three biological replicates (dots), are shown with respect to RLuc-Mlx5′ vector set to a value of 1. Statistical analysis was performed with ordinary analysis of variance (ANOVA) followed by Dunnett's multiple comparison test. **$P < 0.01$, ***$P < 0.001$.

C  Left panel: schematic representation of the luciferase constructs used. The RLuc-Mlx5′ construct and its derived mutants (RLuc-Mlx_{mut}30, RLuc-Mlx Δ75) were transfected in N2a cells in growth conditions together with lnc-SMaRT-expressing plasmids (lnc-SMaRT) or with its derived mutant (lnc-SMaRT_{mut}) in the indicated combinations. Right panel: Luciferase activity data, presented as the mean ± s.e.m. of three biological replicates (dots), are shown with respect to each RLuc control vector (RLuc-Mlx5′, RLuc-Mlx_{mut}30, RLuc-Mlx Δ75) set to a value of 1. Statistical analysis was performed with ordinary analysis of variance (ANOVA) followed by Dunnett's multiple comparison test. ***$P < 0.001$.

Source data are available online for this figure.

mobility compared to the single oligos, suggesting the formation of double-stranded structures. Interestingly, staining with NMM showed a strong decrease of the γ-G4 band (Fig 6A, compare lanes #6 and #10), indicating that the presence of the SMaRT-oligo helped solving the G4 structure of the γ-oligo. In these conditions, a second band resulted positive to NMM (band*). The pairing between γ- and SMaRT- oligos further increased by previous denaturation (Fig 6B, #4 and #5) to the detriment of γ-G4 formation (NMM staining in Fig 6C, #4 and #5). On the contrary, pairing was strongly reduced after stabilization of the G4-structure by KCl (Fig 6B and 6C, #6) as indicated by the increase of both the linear SMaRT-oligo (Fig 6B, #6) and of the γ-G4 (Fig 6C, #6) bands.

In conclusion, these experiments show that the G4 sequence in Mlx-γ can indeed form a G4 structure and that in the presence of the complementary region present in lnc-SMaRT the same region can be engaged to form a duplex.

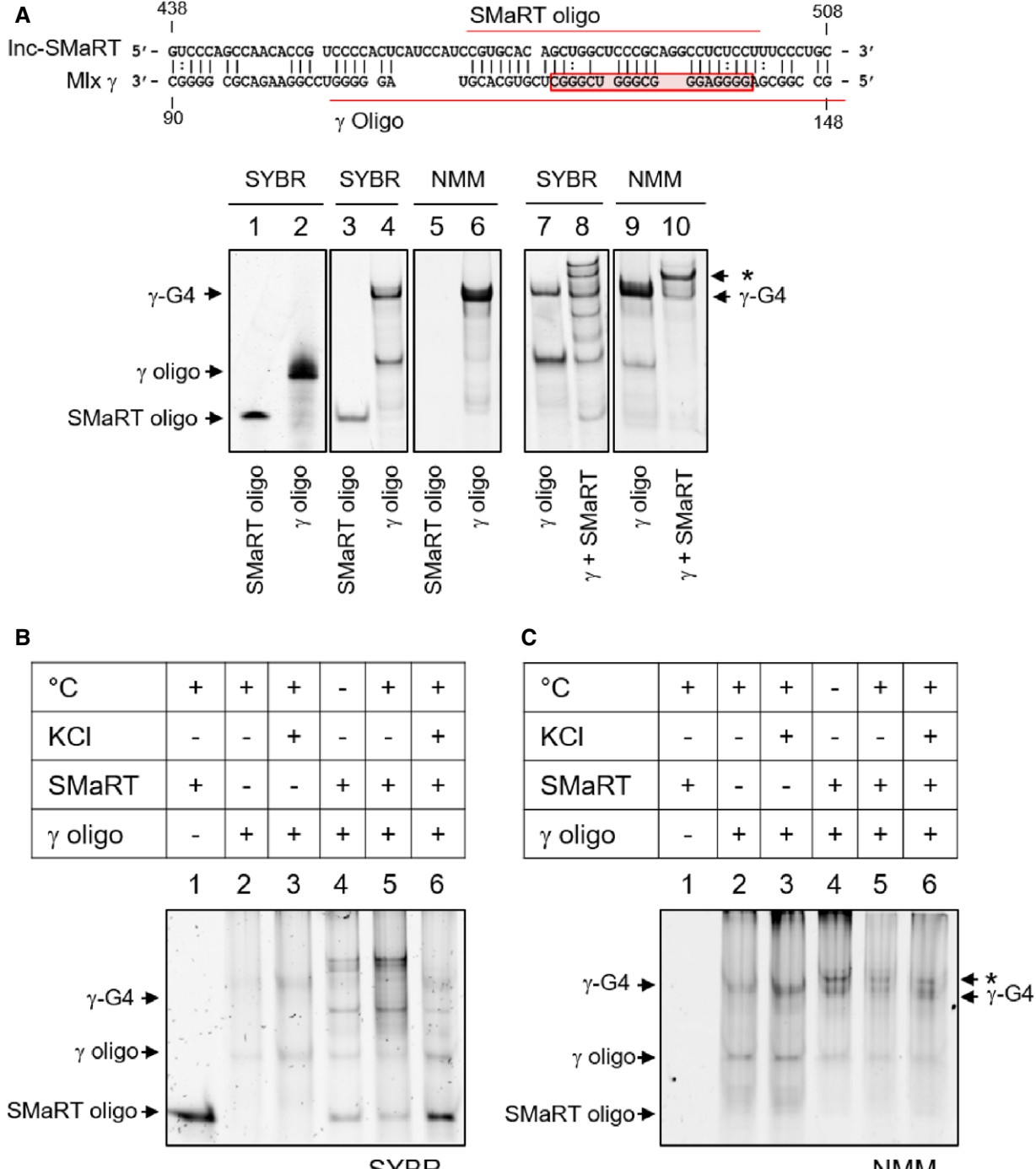

Figure 6.

**Figure 6.** *In vitro* characterization of the interaction between lnc-SMaRT and Mlx-γ pairing sequences.

A   Upper panel: schematic representation of the predicted base pairing between lnc-SMaRT (region A) and Mlx γ mRNA; the predicted G4 structure is boxed in red, while the synthetic oligos sequences are underlined in red. Lower panel: Lanes 1 and 2: 5 pmol of lnc-SMaRT and Mlx synthetic oligos were loaded on a denaturing polyacrylamide gel and stained for total RNA (SYBR GOLD, indicated as SYBR in the figure). Lanes 3 and 4: 5 pmol of lnc-SMaRT and Mlx synthetic oligos were loaded on a native polyacrylamide gel and stained for total RNA (SYBR GOLD). Lanes 5 and 6: 200 pmol of lnc-SMaRT and Mlx synthetic oligos were loaded on a native polyacrylamide gel and stained with the selective G-quadruplex staining N-methyl mesoporphyrin IX (NMM). Lanes 7 and 8: 5 pmol of Mlx oligo alone (7) or in combination with 5pmol of lnc-SMaRT-oligo (8) were heated at 100°C for 10 min and, after slow cooling, loaded on a native polyacrylamide gel, and stained for total RNA (SYBR GOLD). Lanes 9 and 10: 200 pmol of Mlx oligo alone (7) or in combination with 200 pmol of lnc-SMaRT-oligo (8) were heated at 100°C for 10 min and, after slow cooling, loaded on a native polyacrylamide gel and stained with the selective G-quadruplex staining N-methyl mesoporphyrin IX. Asterisk (*) indicates the second NMM-positive band obtained upon SMaRT and γ-oligo interaction.

B   5 pmol of indicated oligos were loaded on a native polyacrylamide gel. Oligos were treated as described. °C = oligos were heated (+) or were not heated (−) at 100°C for 10 min and loaded after slow cooling. KCl = oligos were incubated (+) or were not (−) in a buffer containing 100 mM KCl (for G-quadruplex stabilization). The gel was stained with total RNA staining (SYBR GOLD).

C   200 pmol of indicated oligos treated as in (B) were loaded on a native polyacrylamide gel with the selective G-quadruplex staining N-methyl mesoporphyrin IX. Asterisk (*) indicates the second NMM-positive band obtained upon SMaRT and γ-oligo interaction.

Data information: Representative gels from at least three independent experiments are shown.

## Discussion

Guanine-quadruplexes (G4) consist of non-canonical four-stranded helical arrangements of guanine-rich nucleic acid sequences. In RNA, G4 regions may act in both the nucleus and the cytoplasm, exerting different post-transcriptional effects ranging from RNA processing to localization, stability, and translation [38]. Indeed, G4 sequences were found in quite a high number of transcripts with localization in 5′ and 3′ UTRs as well as in CDS regions [39] and several reports correlated the presence of G4 sequences in 5′UTRs with translational repression [40–42]. Resolution of G4 structures *in vivo* requires specialized enzymes, among them the DHX36 RNA helicase, belonging to the DEAH-box family. This protein exhibits ATP-dependent G4 unwinding activity and it was shown to associate to more than 100 G4-containing RNAs [31]. DHX36 recognizes G4 sequences in 5′ and 3′ UTRs and controls both mRNA translation and stability [43] through its helicase activity; moreover, it can also interplay with microRNA-mediated translational regulation at G4 elements present in 3′UTRs [44]. However, how the activity of the enzyme on these targets is regulated *in vivo* and what are the molecular mechanisms of G4-mediated regulation still remain open questions.

Here, we describe the function of a lncRNA which enters the pathway of the DHX36 helicase and of one of its specific G4-substrates. lnc-SMaRT, by base pairing with the G4-containing region of Mlx-γ mRNA, prevents its translation in an antagonistic manner with DHX36. Lnc-SMaRT is specifically expressed in early phases of myogenesis and elicits an important function since its depletion leads to alteration in the differentiation program with defects in myoblast fusion. Notably, its overexpression produced a clear apoptotic phenotype. In the myogenic lineage, induction of differentiation and apoptosis share a number of cellular mechanisms that involve several caspase family members [45], such as the case of caspase 3 activity required to control correct myotube formation [46]. In this landscape, our data indicate that lnc-SMaRT should be finely controlled in time and quantity in order to fine-tune the balance between differentiation and apoptosis and to ensure proper myogenesis.

Among putative interactors of lnc-SMaRT, we validated the Mlx mRNA by both native and psoralen-crosslinking pull-down conditions. We show that while all three Mlx isoforms (α, β, and γ) share an extended region of complementarity with lnc-SMaRT in their 3′UTR, only Mlx-γ contains a second sequence of complementarity located in the CDS. Notably, this second region includes a G4-element which provides the ability to bind the DHX36 helicase. While the base pairing between the G4-containg element of Mlx-γ and lnc-SMaRT conferred translational control, the 3′UTR-pairing region did not. It is possible that since the luciferase assays were performed in overexpression conditions of both lnc-SMaRT and luciferase constructs, the effect of the 3′UTR-pairing region could have been underscored; therefore, we cannot exclude that in endogenous conditions this pairing could cooperate for a more efficient recognition of lnc-SMaRT for the target mRNA.

MLX proteins were described as myogenic transcriptional factors able to activate a specific class of myokines and to regulate myogenesis in response to glucose signaling [33]. We demonstrated that the three MLX isoforms are part of a regulatory loop where MLX-γ regulates the nuclear re-localization of the other two. In fact, Mlx-γ mRNA downregulation decreased the global levels of MLX proteins inside the nucleus and in turn affected the expression of several target genes. We showed that lnc-SMaRT enters in this circuitry by controlling the translation of MLX-γ and in turn the overall amount of the nuclear MLX proteins. Interestingly, along the myogenic differentiation process, the amount of MLX inside the nucleus inversely correlated with lnc-SMaRT expression, again supporting the inhibitory effect of the lncRNA on MLX-γ synthesis.

In the overall, this circuitry combines the translational control of the MLX-γ isoform by lnc-SMaRT with consequent regulation of the subcellular localization of the total MLX proteins and their transcriptional outcome. This should occur at very specific and restricted stages during *in vitro* differentiation likely ensuring the fine-tuning and robustness to the myogenic process. Indeed, lnc-SMaRT was shown to be present *in vivo* only in dystrophic muscles of *mdx* mice, known to undergo extensive regeneration of muscle fibers. Therefore, it is possible that lnc-SMaRT might control the fine balance between proliferation and differentiation required to control muscle homeostasis during differentiation and regeneration conditions.

The search for putative G-quadruplex forming sequences in lnc-SMaRT with the QGRS Mapper software [32] and with the "QuadBase2" tool [47] did not predict the occurrence of *bona fide* G4-elements. Therefore, it is likely that the interaction of lnc-SMaRT and DHX36 is indirect. One hypothesis is that region B can mediate the initial interaction between lnc-SMaRT and the DHX36/Mlx-γ complex; in turn, the helicase would favor the formation of the pairing between region A and the G4-containing sequence producing the

inhibitory double-stranded structure. This would be in agreement with the finding that DHX36 is able to unwind quadruplex regions and to promote the formation of double-stranded pairing with other RNA sequences [30].

Comparative genomics analysis indicated that the MLX locus is conserved among mammals, starting from Opossum up to human; moreover, in the same species, the Mlx transcript is present in multiple isoforms. Notably, the G4 element present in Mlx-γ also resulted highly conserved, pointing to the important role of these sequences in controlling Mlx production. On the contrary, search for a human counterpart of lnc-SMaRT did not show any conserved synthenic RNA, suggesting that the presence of such sequence in mouse originated from a divergent functional evolution. The region of lnc-SMaRT pairing with the G4 element of Mlx-γ resulted as part of a LINE element, derived from non-LTR retrotransposons [48]. The interesting aspect of the origin, evolution, and functional diversification of lncRNAs is the presence of repetitive sequences, in particular transposable elements (TE), which are "mobile" sequences able to change their position in the genome. Several roles have been attributed to TE in regulating genome evolution, gene expression, genetic instability, and cancer disposition. Notably, the content of TE-derived sequences in lncRNA genes is greater than in protein-coding genes, and it reaches 51% of the mouse lincRNAs [49]. Due to their increased abundance in evolution, it is likely that TE elements have contributed to the evolution and functional diversification of lncRNAs in different species, not only in animals but also in other eukaryotic kingdoms such as plants [50]. Therefore, it is possible that TE spreading has provided an evolutionary drive to increase the RNA–RNA "interactive" potential [51] and that another lncRNA in human may have acquired the ability to control G4 elements.

# Materials and Methods

## Cell culture and treatments

C2C12 murine myoblast (ATCC) were maintained in proliferating conditions in growth medium (Dulbecco's modified Eagle's medium supplemented with 2 mM L-glutamine, 100 U/ml penicillin, 100 μg/ml streptomycin, 20% fetal bovine serum) and induced to differentiate in differentiation medium (FBS reduced to 0.5%). HeLa cells were cultured in Dulbecco's modified Eagle's medium supplemented with 2 mM L-glutamine, 100 U/ml penicillin, 100 μg/ml streptomycin, and 10% fetal bovine serum. N2a cells were cultured in minimum essential medium Eagle (M2279, Sigma), supplemented with 2 mM L-glutamine, sodium pyruvate 1 mM, 1× MEM non-essential amino acid solution (M7145, Sigma), 10% FBS premium USA sourced (45001-106, Corning), 100 U/ml penicillin, and 100 μg/ml streptomycin.

For siRNAs, reverse transfection was performed as follows: For a 3.5-cm culture dish, 5 μl of Lipofectamine RNAiMAX (Thermo Scientific) was added to 300 μl of Opti-MEM® I reduced serum medium (Gibco); the siRNA was then added at a final concentration of 30 nM for a final volume of 2 ml. After 15 min of incubation at room temperature, the transfection mix was distributed on the culture dish and 200,000 C2C12 cells were seeded in GM. After 24 h, cell confluency was checked and the culture medium was replaced with DM, in order to trigger the differentiation; cells were collected 48 h after the induction. Scramble and DHX36 siRNA were purchased from Qiagen (AllStars Negative Control siRNA and Mm Dhx36-4 FlexiTube, respectively), while lnc-SMaRT siRNAs were custom-synthetized (Qiagen). Plasmid DNA transfection was performed using Lipofectamine 2000 (Thermo Fisher Scientific) according to the manufacturer's specifications.

## RNA extraction and analysis

RNA extraction was performed with the Direct-zol Miniprep RNA Purification Kit (Zymo Research) with on-column DNase treatment, according to the manufacturer's instructions. For RNA pull-down and RIP experiments, the RNA was extracted using QIAzol reagent and miRNEasy spin columns (QIAGEN), according to the manufacturer's specifications.

For routine experiments, total RNA was retro-transcribed with PrimeScript™ RT Reagent Kit (Takara) according to the manufacturer's instructions while, for low RNA input experiments (RNA pull-down and RIP), the Superscript VILO cDNA Synthesis Kit was used (Life Technologies). Samples were then analyzed by qPCR using PowerUp SYBR Green Master Mix (Thermo Fisher Scientific) or by semi-quantitative RT–PCR using MyTaq™ DNA polymerase (Bioline). The oligonucleotide sequences are listed in Table EV4.

## Protein extraction and Western blot

Protein extract was obtained using standard RIPA buffer, supplied with 1× Complete Protease Inhibitor Cocktail (Roche). Protein concentration was assessed using Bradford Protein Assay (Bio-Rad). Protein electrophoresis was performed using 4–15% Mini-PROTEAN TGX Precast acrylamide gel (Bio-Rad) according to the manufacturer's instructions, and proteins were transferred to Immobilon-E PVDF 0.45-μm membrane (Merck-Millipore) at 80 V for 1 h in 1× Towbin Transfer Buffer (25 mM TRIS, 192 mM glycine, 20% methanol). Membranes were blocked with 5% non-fat dry milk (Difco skim milk) for 1 h and incubated overnight at 4°C with the following primary antibodies: anti-DHX36 (13159-1-AP, Proteintech); anti-MLX (12042-1-AP, Proteintech); anti-MyoD (M-318, Sc-760, Santa Cruz Biotechnology); anti-MYOG (F5D, sc-12732, Santa Cruz Biotechnology); anti-caspase 3 (9622, Cell Signaling Technology); anti-ACTININ-4 (G-4, sc-390205, Santa Cruz Biotechnology); anti-DYS (NCL-DYS1, Novocastra Laboratories), anti-MEF2C (sc-365862, Santa Cruz Biotechnology), anti-GAPDH (6C5, sc-32233, Santa Cruz Biotechnology), and anti-HPRT (FL-218, sc-20975, Santa Cruz Biotechnology). The following secondary antibodies were used: goat anti-rabbit HRP (31460, Invitrogen) and goat anti-mouse HRP (32430, Invitrogen). FLAG-tagged proteins were detected with ANTI-FLAG® M2-Peroxidase (HRP) antibody (A8592, Sigma). Protein detection was carried out with WesternBright ECL (Advansta) using ChemiDoc™ MP System and images were analyzed using Image Lab™ Software (Bio-Rad).

## Native RNA pull-down

Native pull-down on C2C12 myoblasts was performed as described in ref. [19]. The sequences of the biotinylated oligonucleotides belonging to probe set-1, probe set-2, and LacZ control are listed in Table EV4.

## Protein mass spectrometry

To identify unknown protein partners of lnc-SMaRT, protein mass spectrometry after purification of RNA–protein complexes was used. Eluted proteins from the lnc-SMaRT pull-down were loaded on a 4–12% NuPAGE® Bis-Tris gels (Life Technologies); then, the corresponding gel parts of each sample were sent for mass spectrometry analysis to the proteomic platform at IGBMC–Strasbourg, where they were digested with trypsin and analyzed according to their standard mass spectrometry pipeline using LTQ Velos Pro instrument (Thermo Fisher). To analyze the results, stringent filtration criteria were applied (1% false discovery rate and 2 peptides minimum per protein).

## DHX36 RNA immunoprecipitation

Two 10-cm plates of C2C12 ($1.5 \times 10^6$) differentiated for 48 h were collected in PLB Buffer (KCl 100 mM; MgCl$_2$ 5 mM; NP-40 0,5%; DTT 1 mM; protease and Rnase inhibitor), incubated for 15 min at 4°C on a rotating wheel and then centrifuged at 15,000 $g$ for 10 min at 4°C. The supernatant was recovered and protein concentration was quantified by Bradford assay; 0.5 mg of extract was used for each sample (IP and IgG) and was pre-cleared with 40 µl of Protein G Agarose/Salmon Sperm Beads (Millipore) in a final volume of 1 ml of NT2 buffer (Tris–HCl pH 7.4 50 mM; NaCl 150 mM; MgCl$_2$ 1 mM; NP-40 0.05%; Protease and RNase Inhibitor) for 2 h at 4°C on a rotating wheel. 10% of the final volume was taken as input, and the remaining pre-cleared lysate was incubated with 5 ug of DHX36 (Proteintech, 13159-1-AP) or IgG (Santa Cruz Biotechnology, sc-2027) antibodies overnight at 4°C. Subsequently, 80 µl of protein G agarose beads were added to each sample and incubated for 2 h at 4°C. After antibody–protein complex recovery, the beads were washed 4× in NT2 buffer and finally re-suspended in 200 µl of NT2 Buffer. To check DHX36 immunoprecipitation efficiency, an aliquot of 50 µl of beads was pelleted, re-suspended in 1× Laemmli sample buffer (Bio-Rad) and 50 mM DTT, and incubated at 70°C for 15 min; then, the eluate was used for Western blot.

RNA was recovered from the remaining 150 µl of beads by re-suspending them in 500 µl of TRI-Reagent (Zymo Research). RNA extraction was carried out with RNeasy Plus Minikit (Qiagen) according to the manufacturer's protocol.

## Psoralen-crosslinked RNA pull-down

Psoralen-crosslinked RNA pull-down protocol was adapted from RICC-seq protocol [52]. C2C12 cells were plated in two 10-cm plates at a density of $1 \times 10^6$ cell/plate and grown for 24 h before switching to DM. After 48 h of differentiation, the medium was replaced with fresh DM supplemented with 20 µg/ml of 4′-aminomethyl-4,5′,8-trimethylpsoralen (AMT, Cayman Chemical) and cells were incubated 30 min at 37°C. After incubation, the cells were washed twice with complete PBS, covered with PBS supplemented with 150 µg/ml AMT, and crosslinked at 365 nm at 10-min intervals for 1 h. Cells were lysed in 500 µl of guanidinium hydrochloride 3M, and the lysate was subdivided into 150 µl aliquots; 25 µl of a 20 mg/ml solution of RNAse-free proteinase K (Ambion) and 7.5 µl of 20% sodium dodecyl sulfate (SDS) were added to each aliquot; and the samples were incubated at 65°C for 1 h with gentle agitation. RNA was precipitated with conventional phenol/chloroform extraction; the rest of pull-down protocol was performed as described in [52], with minor modifications.

## Microscopy and image analysis

C2C12 cells were cultured on pre-coated glass coverslips (300 ug/ml in PBS Collagen Rat Tail, Corning) and then were fixed in 4% paraformaldehyde (Electron Microscopy Sciences, Hatfield, PA) in PBS at 4°C for 20 min.

MHC immunofluorescence was performed as previously described [53]. Samples were imaged on inverted microscope Zeiss AxioObserver A1 equipped with Axiocam MRM R camera and Plan-Neofluar EC 10×/0.3 M27 objective. Images were acquired with AxioVision Rel.4.8 software. At least five randomly chosen microscope fields of two independent biological samples were analyzed ($n > 600$ cells for each field). For MLX and MHC double immunostaining, cells were permeabilized with Triton 0.2% for 10 min, blocked with 2% BSA/PBS for 20 min, and subsequently incubated at 4°C overnight with anti-MLX (Proteintech, 12042-1-AP) diluted 1:50 and with anti MHC (MF20 clone hybridoma supernatants) diluted 1:2 in blocking solution. After serial washes in 0.1% Triton/PBS, coverslips were incubated in 1% goat serum/1% donkey serum/PBS with goat anti-rabbit Cy3 conjugated (1:300; Jackson ImmunoResearch) and donkey anti-mouse AlexaFluor 488 (1:200, Life Technologies) to detect MLX and MHC primary antibodies, respectively. The incubations were performed for 1h at room temperature. The specificity of immunolabeling was verified in control samples prepared with the incubation buffer alone, followed by the secondary conjugated antibody. The nuclei were stained with DAPI (4′,6-diamidino-2-phenylindole).

Confocal images (16-bit and 1,024 × 1,024 pixels) were acquired with 60× NA 1.35 oil objective (UPLANSApo) on inverted microscope (Olympus IX73) equipped with a Confocal Imager (CREST X-LIGHT) spinning disk, a CoolSNAP Myo CCD camera (Photometrics), and a Lumencor Spectra X LED illumination. The Z-stacks were collected at step size of 0.2 µm with XY resolution of 0.075 µm.

In post-acquisition processing, a specific range of intensity balance was manually determined by using MetaMorph or FIJI software to the entire image. A qualitative display of MLX distribution inside the cells was performed with FIJI software by threshold processing (to obtain binary images) and local maxima filter (to mark the peaks of MLX signals inside the nuclei).

The Nuc/Cyt fluorescence ratio (fluorescence signal intensity inside nucleus/fluorescence signal intensity inside cytoplasm) of MLX signals was obtained on single-focus Z-plane by measuring the mean fluorescent intensity inside composite selections (ROI) that delimitate subcellular compartments. Nuclear boundary was assigned by DAPI staining, while the edge of the cells was highlight by MHC staining. Nuc/Cyt ratio value = 1 indicates the 1:1 fluorescent ratio of MLX signals between nuclear and cytoplasmic compartment, while Nuc/Cyt values > 1 or < 1 indicate the enrichment of the MLX signals inside the nucleus or cytoplasm, respectively.

The fluorescence was quantified using FIJI software from about 50 cells on each condition (20 cells on each condition for rescue

analysis shown in Fig EV4B). The value of the Nuc/Cyt ratio was represented as mean of two biological replicates.

Acquisition and quantification of myotubes formation (mononucleated MHC-positive cells and myotube Fusion Index) were performed as described in ref. [54].

## lnc-SMaRT overexpression constructs

The construct for the overexpression of lnc-SMaRT was obtained by cloning lnc-SMaRT cDNA in a modified pCDNA3.1(+) plasmid (Invitrogen), in which CMV promoter has been replaced with the human EIF1a promoter sequence, using XhoI and NotI restriction enzymes (Thermo Scientific). The lnc-SMaRT$_{mut}$ mutant was obtained by inverse PCR with divergent primers from the lnc-SMaRT overexpressing plasmid. Oligonucleotides are listed in Table EV4.

## FLAG-MLX isoforms overexpression constructs

The 5′UTR and coding sequence of the three MLX isoforms were amplified from C2C12 D2 cDNA with the MLX Infusion Fw and Rv oligonucleotides. The obtained PCR products were inserted in the previously described pCDNA3.1-EIF1a backbone, linearized with XhoI and NotI restriction enzymes (Thermo Scientific), using the In-Fusion HD Cloning Kit (Clontech).

FLAG tag was in-frame inserted at the N-terminus by inverse PCR using MLX FLAG Fw and Rv primers.

## lnc-SMaRT overexpressing stable cell line

An expression cassette containing the EIF1a promoter, the lnc-SMaRT cDNA sequence, and the BGH termination site has been amplified from the already described lnc-SMaRT overexpression plasmid. The cassette was subsequently inserted in the ePB-BSD-PGK-Int-RFP [55] linearized with XhoI-BglII restriction enzymes (Thermo Scientific) with the In-Fusion HD Cloning Kit (Clontech). To generate the stable lnc-SMaRT overexpressing cell line, 200,000 C2C12 cells were transfected with 5 μg of SMaRT-ePB-BSD-PGK-Int-RFP and 500 ng of the *piggyBac* transposase vector. Cells were selected in 10 μg/ml of blasticidin S (Thermo Scientific) for 7 days and then tested for genomic integration of the cassette and consistent overexpression of the full-length lnc-SMaRT transcript.

## TUNEL assay

Apoptotic quantification was performed by using *In Situ* Cell Death Detection Kit, Fluorescein (Roche 11684795910) according to the manufacturer's instructions with minor modifications. Briefly, PFA-fixed-differentiated C2C12 cells were permeabilized with 0.3% Triton X-100/PBS on ice for 15 min. After extensive washing with PBS, cells were incubated with TUNEL reaction mixture (Enzyme solution diluted 1:10 with Label Fluorescein solution) at 37°C for 2 h in Top Brite automatic slide hybridizer (Resnova). Following two washes with PBS, DAPI solution (1 μg/ml) was added at room temperature for 5 min. Finally, samples were mounted with ProLong Diamond Antifade Mountant (ThermoFisher Scientific P-36961), and the images were acquired using a LUCPlanFLN 20× objective (NA 0.45) and a UPLANSApo 60× oil objective (NA 1.35) and collected as described in "image analysis" section. The

apoptotic rate indicates the percentage of apoptotic cells with respect to the total number of nuclei; it was quantified by manual counting of Fluorescein-labeled nuclei (with DNA strand breaks generated during apoptosis) and DAPI-labeled nuclei in a microscope field. The value of the apoptotic rate was represented as mean ± SEM of two biological replicates.

## Luciferase Reporter constructs

The RLuc-Mlx 5′ construct was obtained by cloning the exon 1 of MLX γ in Ψcheck2 vector (Promega) upstream the Renilla luciferase coding sequence previously depleted of its start codon by inverse PCR (see the scheme in Fig 4A); the In-Fusion HD Cloning Kit (Clontech) was used. The RLuc-Mlx 5′–3′ and RLuc-Mlx 3′ reporter constructs were obtained cloning the MLX 3′UTR sequence in the RLuc-Mlx 5′ or in the Ψcheck2 vector (Promega) using XhoI and NotI restriction enzymes (Thermo Scientific).

In the RLuc-Mlx Δ75 mutant, a deletion of 75 bp was obtained by inverse PCR with divergent primers from the full-length RLuc-Mlx 5′ construct. The RLuc-Mlx$_{mut}$30 mutant was generated by inverse PCR with divergent primers.

Oligonucleotides are listed in Table EV4.

## Luciferase assays

C2C12, HeLa, or N2a cells were transiently transfected with the luciferase reporter plasmids in the indicated combinations, using Lipofectamine-2000 Reagent (Thermo Scientific). The reporter plasmids also contain the Firefly luciferase (FLuc) gene to normalize for transfection efficiency. 48 h after transfection, cells were lysed, and RLuc and FLuc activities were measured by Dual Glo Luciferase assay (Promega). Transfection of each construct was performed in triplicate as well as Luciferase assays. Ratios of RLuc readings to FLuc readings were taken for each experiment, and triplicates were averaged.

## In-gel G-quadruplex staining

G-quadruplex formation and in-gel staining were carried out as described in ref. [30]. The oligonucleotide sequences are listed in Table EV4.

## RNA sequencing and bioinformatics analyses

TruSeq Stranded Total RNA Library Prep Kit with Ribo-Zero Gold was used to prepare cDNA libraries for both lnc-SMaRT depletion and pull-down RNA-Seq experiments. The sequencing reactions, performed on an Illumina Hiseq 2500 Sequencing system at the Institute of Applied Genomics (IGA; Udine, Italy), produced an average of 21.6 million 100 nucleotide long paired-end read pairs per sample for the depletion and an average of 55.8 million 50 nucleotide long single-end reads per sample for the lnc-SMaRT pull-down experiment. Reads from both experiments were pre-processed using Trimmomatic software [56] which removed adapter sequences and poor quality bases, filtering out those reads whose length after trimming was less than 30 nucleotides. Then, Bowtie 2 [57] was used to identify and discard reads mapping to rRNAs and tRNAs. In order to calculate the distribution of the inner distance between mate pairs,

reads from depletion experiment were aligned to a non-redundant set of murine RNA sequences derived from Ensembl 77 mouse annotation [58] using BWA software [59]. We estimated mean and variance of the inner distance distribution from aligned read pairs whose inner distance was within interval [Q1-2(Q3-Q1),Q3 + 2(Q3-Q1)] (Q1 = first quartile, Q3 = third quartile). TopHat2 [60] was employed to align reads to GRCm38 mouse genome and Ensembl 77 transcriptome using parameters -i 50 -r 40 –mate-std-dev 80 –library-type fr-firststrand for depletion and -i 50 –library-type fr-first-strand for pull-down experiment. Reads mapping to mitochondrial genome were filtered out. Read numbers and mapping statistics are reported in Table EV2 (lnc-SMaRT depletion) and Table EV3 (lnc-SMaRT pull-down).

For the depletion experiment, we used Htseq-count software in intersection-strict mode [61] to count reads mapping to Ensembl 77 genes. edgeR R package [62] was used to perform differential gene expression analysis after filtering out genes with a CPM (Count Per Million) value less than 1 in at least two samples. TMM and GLM robust normalization were applied to read counts. Model fitting and testing was performed using glmFIT and glmLRT functions. Differentially expressed genes were selected using an FDR cutoff of 0.05. Heatmap of differentially expressed protein-coding genes was drawn based on $\log_2$-transformed RPKM values, calculated using the edgeR rpkm function. MDS plot was drawn using edgeR plotMDS function. Gene ontology biological process term enrichment analysis was performed using DAVID Functional Annotation Tool [63] on protein-coding genes upregulated and downregulated after lnc-SMaRT depletion, using all protein-coding genes tested for differential expression as background. The GOsummaries R package [64] was used to create a word cloud for the enriched GO terms (Benjamini–Hochberg-corrected $P$-value < 0.05).

Aligned reads from pull-down experiment were further processed using Picard MarkDuplicates (available at http://broadinstitute.github.io/picard) to remove duplicate reads and Bamtools [65] to filter out reads mapping to multiple positions.

Piranha tool [66] was used to call peaks for lnc-SMaRT pull-down and LacZ pull-down reads, using Input reads as a covariate (lnc-SMaRT_VS_Input and LacZ_VS_Input, respectively); bin size was set to 200. Peaks were also called for lnc-SMaRT pull-down versus LacZ pull-down (lnc-SmaRT_VS_LacZ). Transcripts bound by lnc-SMaRT were selected as those whose exons overlapped peaks from both lnc-SmaRT_VS_Input and lnc-SmaRT_VS_LacZ and did not overlap peaks from LacZ_VS_Input. BEDtools intersect tool [67] was used to evaluate these overlaps.

RNA–RNA interaction prediction was computed using IntaRNA 2.3.0 [68] and with -n 3 setting. Graphic representations of IntaRNA interactions between lnc-SMaRT and MLX were computed using Forna [69]. C++ implementation of QGRS mapping algorithm [32] with -t 3 –s 35 settings was used for G-quadruplexes predictions.

Evolutionary conservation of specific Mlx transcript isoforms was based on the analysis of intron-spanning reads from RNA-Seq data retrieved from Ensembl [70] database.

**Statistical analyses**

Unless stated otherwise, data are shown as mean ± SEM; the number of biological replicates is indicated in each Figure legend.

Scatter-and-bar plots have been used to show individual biological replicate values. Statistical tests used to assess significance of differences between means are indicated in each Figure legend. $P$-values below 0.05 were marked by 1 asterisk, while 2 asterisks indicate a $P$-value < 0.01 and 3 asterisks a $P$-value < 0.001. $P$-values < 0.05 were considered significant.

## Data availability

The RNA-Seq data from this publication have been deposited to the GEO database (https://www.ncbi.nlm.nih.gov/geo/) and assigned the identifier GSE128486.

**Expanded View** for this article is available online.

## Acknowledgments

The authors acknowledge O. Sthandier and M. Marchioni for technical help and M. Arceci for experimental support. S.S. was recipient of a EC—FP7 Marie Curie (ITN - 2013 GA 607720). This work was partially supported by grants from ERC-2013 (AdG 340172–MUNCODD), H2020 - ERC-2019-SyG (855923-ASTRA), Telethon (GGP16213), AIRC (IG 2019 Id. 23053), and PRIN 2017 (2017P352Z4) to I.B. The authors wish to dedicate this manuscript to all people in healthcare organizations operating around the world to deal with COVID-19 infections.

## Author contributions

Conceptualization: IB; methodology: IB, JM, and DM; formal analysis: AS, AC, and AP; investigation: JM, DM, SS, TS, FC, DD, and MM; data curation: AC; writing—original draft: IB and JM; visualization: JM and DM; funding acquisition: IB; resources: IB; and supervision, IB.

## Conflict of interests

The authors declare that they have no conflict of interest.

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
