## [Review Process File · EMBO Reports]

SMaRT lncRNA controls translation of a G-quadruplex containing mRNA antagonizing the DHX36 helicase

Julie Martone, Davide Mariani, Tiziana Santini, Adriano Setti, Sama Shamloo, Alessio Colantoni, Francesca Capparelli, Alessandro Paiardini, Dacia Dimartino, Mariangela Morlando, Irene Bozzoni

Review timeline:

Submission date:	23 December 2019
Editorial Decision:	27 January 2020
Revision received:	25 February 2020
Editorial Decision:	25 March 2020
Revision received:	26 March 2020
Accepted:	31 March 2020

Editor: Esther Schnapp

Transaction Report:

1st Editorial Decision

27 January 2020

Thank you for the transfer of your manuscript to EMBO reports. As discussed, I sent your study to 3 new referees and we have now received the full set of referee reports that is pasted below.

As you will see, all referees acknowledge that the findings are potentially interesting. However, they also point out that the data need to be strengthened, especially in terms of statistical reporting, data presentation and physiological relevance (eg regarding SMaRT lncRNA expression levels). I think that all referee comments are sensible and should be addressed. Please let me know if you disagree.

I would thus like to invite you to revise your manuscript with the understanding that the referee concerns must be fully addressed and their suggestions taken on board. Please address all referee concerns in a complete point-by-point response. Acceptance of the manuscript will depend on a positive outcome of a second round of review. It is EMBO reports policy to allow a single round of major revision only and acceptance or rejection of the manuscript will therefore depend on the completeness of your responses included in the next, final version of the manuscript.

Revised manuscripts should be submitted within three months of a request for revision; they will otherwise be treated as new submissions. Please contact us if a 3-months time frame is not sufficient for the revisions so that we can discuss this further. You can either publish the study as a short report or as a full article. For short reports, the revised manuscript should not exceed 27,000 characters (including spaces but excluding materials & methods and references) and 5 main plus 5 expanded view figures. The results and discussion sections must further be combined, which will help to shorten the manuscript text by eliminating some redundancy that is inevitable when discussing the same experiments twice. For a normal article there are no length limitations, but it should have more than 5 main figures and the results and discussion sections must be separate. In both cases, the entire materials and methods must be included in the main manuscript file.

Regarding data quantification, please specify the number "n" for how many independent

experiments were performed, the bars and error bars (e.g. SEM, SD) and the test used to calculate p-values in the respective figure legends. This information must be provided in the figure legends. Please also include scale bars in all microscopy images.

- 1) A data availability section providing access to data deposited in public databases is missing
- 2) Your manuscript contains statistics and error bars based on n=2 or on technical replicates. Please use scatter blots in these cases. No statistics can be calculated if n=2.

2) individual production quality figure files as .eps, .tif, .jpg (one file per figure).

See https://wol-prod-cdn.literatumonline.com/pb-assets/embo-site/EMBOPress_Figure_Guidelines_061115-1561436025777.pdf for more info on how to prepare your figures.

5) a complete author checklist, which you can download from our author guidelines <https://www.embopress.org/page/journal/14693178/authorguide>. Please insert information in the checklist that is also reflected in the manuscript. The completed author checklist will also be part of the RPF.

6) Please note that all corresponding authors are required to supply an ORCID ID for their name upon submission of a revised manuscript (<https://orcid.org/>). Please find instructions on how to link your ORCID ID to your account in our manuscript tracking system in our Author guidelines <https://www.embopress.org/page/journal/14693178/authorguide#authorshipguidelines>

8) We would also encourage you to include the source data for figure panels that show essential data. Numerical data should be provided as individual .xls or .csv files (including a tab describing the data). For blots or microscopy, uncropped images should be submitted (using a zip archive if multiple images need to be supplied for one panel). Additional information on source data and instruction on how to label the files are available at <https://www.embopress.org/page/journal/14693178/authorguide#sourcedata>.

I look forward to seeing a revised version of your manuscript when it is ready. Please let me know if you have questions or comments regarding the revision.

REFeree REPORTS

Referee #1:

The Martone et al manuscript describes the long non coding RNA lnc-SMART that inhibits translation of G-quadruplex containing MLX-gamma mRNA by direct base-pairing with the G4-containing region of Mlx-gamma mRNA. This inhibition occurs by counteracting the effect of RNA helicase DHX36 that the authors identified as one of the lnc-SMART binding partners (likely indirect). As MLX proteins are myogenic transcription factors, which are tightly regulated during myogenesis. The authors show that the regulation of MLX protein translation and subcellular localization is dependent on lnc-SMART. Mis-expression of lnc-SMART by its knockdown or overexpression has a profound effect on normal differentiation of C2C12 cells. While the interaction of lncSMART with both the G4-region of Mlx-gamma RNA and DHX36 RNA helicase was identified by the authors as a rather educated guess from their RNA pull-down experiments, the study is well-done and the main findings are well put together in one coherent story that merits publication in EMBO reports. My clarification requests and technical issues are listed below.

Major concerns:

1. My major request to the authors is that all measurements of transcript levels should be done by quantitative RT-PCR using at least three biological replicates (which is standard in the field). Data presented in the manuscript are often non-quantitative, lacking proper normalizations and controls and therefore are hard to interpret. In graph plots, it is not explained if biological or technical replicates are shown, which should be indicated in figure legends. In addition, Figure 1E and 3A

should indicate the number of independent biological samples and individual cells analysed.

2. Same for Fig2B, C (pull down and RIP) should show quantification with qRT-PCR rather than non-quantitative bands on the gel. The specificity of the lncSMART-DHX36 interaction should be demonstrated by testing additional lncRNAs expressed at the levels similar to lnc-SMART.

3. The authors should calculate and mention the approximate number of lnc-SMART molecules in C2C12 cells and in vivo. This information is key to understand the molecular mechanism of lnc-SMART especially because of its repressive effect on Mlx-gamma translation by direct base-pairing with its G4 region (which I guess happens stoichiometrically?) and sub-cellular localization of MLX proteins. This point should be clarified in the text. In addition, it would be very informative to clarify the specificity of lncSMART pairing to the G4 region of Mlx-gamma mRNA i.e. would lncSMART inhibit translation of any RNA containing G4 elements in the 5'UTR (the authors mention that more than 100 G4-containing RNAs are known) or is the effect specific to Mlx-gamma only?

Minor comments:

4. The authors should indicate the expression levels of the lnc-SMART relative to MyoD and myogenin.
5. Based on data presented in Fig 1 B, Fig EV1A, there is only one isoform lnc-SMART. The authors should mention if there are additional lncRNA isoforms or it is always only one mature lnc-SMART in C2C12; the size of the main mature lnc-SMART isoform should be mentioned in the text.
6. "In vivo, lnc-SMaRT is present in conditions of muscle regeneration (Fig. EV1B), exemplified by skeletal muscles of dystrophic mdx mice". Does it mean that lncSMART is not expressed in the muscle under homeostatic conditions? This is a very interesting point and should be better explained in the main text.
7. The authors should indicate the location of siRNAs they used for lncRNA-SMART KD in Figure 1A as well as the location of qPCR primers used in Figure 1D.
8. Is downregulation of late differentiation markers shown in Fig EV1C significant? It should be indicated.
9. GO analyses in Figure 1F should be presented in a form of a heatmap or a table with p-values indicated for each term; from the current data presentation one cannot easily see ranking/p-values of all GO terms.
10. Because the authors use siRNA to KD lnc-SMART which has both nuclear and cytoplasmic localization, the KD levels in the nucleus and cytoplasm should be shown as done in figure 1B
11. I would suggest to change "a cDNA copy of lnc-SMaRT" in the first part of the results section to "mature lnc-SMART amplified from cDNA". What promoter was used for mature lncRNA overexpression?
12. In Fig EV2C, when showing sequence complementarity between Mlx and lncSMART transcripts, actual sequences should be shown (as supplementary figure), which are more informative than just a cartoon which is misleading as it shows complementarity of only a few nucleotides.
13. As already mentioned above, the anti-DHX36 RIP experiment in Fig 2E should be presented as a qRT-PCR and not as gel bands. Appropriate controls including DHX36 interaction with other lncRNAs should be included.
14. The numbering of Fig EV2B and Fig EV2C is mixed up and should be fixed.
15. Western blot showing down-regulation of MLX-gamma upon lnc-SMART overexpression in HeLa cells should be quantified (Fig 2F). The number of independent replicates should be indicated.
16. In the left panel of Fig 4A, the constructs are not aligned with the results in the right panel, making the figure very confusing.
17. Fig 4B is not described in the text.

Referee #2:

Martone and colleagues describe a lncRNA, named SMaRT, that is expressed in early phases of myoblast differentiation. Knockdown of SMaRT in C2C12 cells impairs myoblast differentiation. SMaRT is bound by DHX36, a protein involved in unwinding G4 structures. Among transcripts bound by SMaRT, the authors identify mRNAs with predicted G4 structures including MLX. Changes in SMaRT level affect MLX translation without impacting mRNA levels, in a DHX36

dependent manner. The authors propose SMaRT binds Mlx-g and impacts its translation and the nuclear localization of MLX proteins and transcriptional output. While the work is interesting further experiments to support the conclusions.

1- I have a general concern with data representation in the paper. Bar plots do not provide a full sense of the variability between the individual replicates and should be replaced by the actual individual values. A combination of bars and points would be suitable but as described elsewhere (Beyond Bar and Line Graphs: Time for a New Data Presentation Paradigm Tracey L. Weissgerber, Natasa M. Milic, Stacey J. Winham, Vesna D. Garovic Published: April 22, 2015 <https://doi.org/10.1371/journal.pbio.1002128>) replacing data points with bars representing SE or SD is not a faithful representation of the data. Related, instead of representative experiments, the data for the individual biological replicates should be provided as this gives a better sense of the actual variability between experiments. This might already be the case in some experiments but this is not specified and legends refer to "results of representative experiments". For western blots this could be achieved by quantifying the intensity. Uncropped blots don't seem to have been made available.

2- Do changes in Mef2C, Mck and Dys mRNA translate into changes in protein levels? Given the phenotype is unlikely to be driven by changes in mRNA this data is important.

3- Detailed information on the RNAseq data is missing. For example do gene expression differences between samples allow to distinguish siRNA and control treated samples (PCA)? What is the general extent of changes and significance (volcano plot)? What is the actual enrichment in different GO classes and their significance? The way the data is currently presented does not allow the reader to quantitatively assess the results.

4- To validate their siRNA results, the authors establish a C2C12 line overexpressing SMaRT (page3). The authors justify their experiments based on the low transfection efficiency of C2C12. While this is likely to be a problem, overexpression does not directly address the concern and cannot be used as validation. Notably while the levels of MyoD are not affected by KD they seem to be affected by overexpression. Generally, are the levels of affected genes anticorrelated between KD and OE cells? It's also unclear why proliferation and apoptosis is assessed in these cells but not in cells where SMaRT is KD. Given the siRNA transfection is efficient wouldn't OE of a siRNA resistant SMaRT be a better experiment.

5- Related to the above point, if C2C12 transfect inefficiently how is this accounted for in the reporter experiments reported in Figure 4? Would this explain the relatively small effect sizes reported for the analysis reported in Fig 4?

6- Analysis of the RNA pull-down experiments should be more extensive (in line with comments for RNAseq) and made available.

7- The control RNA used in RNA- pull down (LacZ) has a very different GC content from that of endogenous mouse mRNAs. The impact on this on the pull-down results should be discussed.

8- The available analysis of the RNA recovered after SMaRT pull down is too superficial. Are mRNAs bound by the lncRNA part of a specific GO class? Do they have skewed GC contents? Length?

9- Of all the mRNAs that putatively interact with SMaRT, how many have a predicted G4? Is 18 more than expected by chance? Are the confidence scores of G4 predictions within this 18 mRNAs higher than for other mRNAs? Are these G4 also predicted by other methods?

10- Is there a G4 sequence in SMaRT?

11- What happens to Mlx-g translation and MLX localization in C2C12 SMaRT overexpressing cells?

12- Are MLX target genes enriched among genes differentially expressed upon SMaRT knockdown?

13- Addition of DHX36 should impact interaction between G4-Mlx-g and complementary region in SMaRT (Figure 4).

Other comments

14- Since SMaRT appears to function in a G4 dependent manner, the topic is not covered in the introduction. It is mentioned in the discussion but to understand the results the reader needs to know what G4 are, the proteins involved in their resolution and examples of other G4-related lncRNAs.

Minor Comments

15- It would be useful to mention early on that SMaRT was identified in mouse.

Referee #3:

The work by Martone and co-workers shed light on the role of the regulatory lncRNA SMaRT, which is able to form G4-mediated RNA duplexes with target mRNAs. The MS is generally very well written and it is easy to follow albeit the high methodology complexity and the model proposed. Shortly, the SMaRT noncoding transcript was first identified in the context of myogenesis. The authors found that SMaRT can directly interact with target mRNAs forming duplexes and modulating translation. This finding is based on a SMaRT-pull down assay using independent interspersed sets of biotinylated probes, and compared to an anti-LacZ set of primers used as a negative control. SMaRT-pulldown was followed by sequencing for associated RNAs identification. In particular, SMaRT can recognize *in vivo* three alternative isoforms of the Mlx mRNA. However, SMaRT can only impair the translation of the Mlx-y isoform, with which it can interact with an additional region present in the mRNA forming a G4 complex. Furthermore, the authors performed a SMaRT-pulldown followed by protein precipitation and MS. This approach allowed the identification of DHX36 as a potential interactor, an enzyme capable of recognizing and unwinding G-quadruplex structures. The protein-RNA interaction was accurately confirmed by alternative methods, e.g. RIP using specific antibodies. Although G4 RNA structures were previously associated to several molecular mechanisms including translational modulation, in this work Martone and co-authors show a very compelling case of an inter-RNA G4 structure mediating the lncRNA regulation over its target mRNA counterpart. Moreover, the authors shed light on the impact of SMaRT-MLX-y interaction over translation of specific mRNA isoforms and the sub-cellular distribution of the translated Mlx pool of proteins. In this part, I should say that while reading the MS I had doubts about the conclusions drawn by the authors based on the FLAG fusions. I first thought that if the FLAG tag was fused to the end of the gene, then truncated FLAG-free proteins may be produced. However, reading carefully the M&M section I found out that the FLAG was fused to the 5' end of the gene, discarding this possibility. I suggest the authors to include a phrase in the figure legend or scheme in the figure itself explaining the fusion construct. I'd like to suggest a few more points to improve the MS:

Major point:

Although the rationale of the MS is easy to follow and the order of the experiments is logical, I find that the first results could be strengthened based on the later ones. In particular, Figure 2F shows the effect of SMaRT over Mlx translation. Later in the MS the authors characterize what part of the lncRNA matches its target mRNA using a very elegant reporter-based approach and *in vitro* assays. I wonder why the authors didn't try to over express the different mutant versions of SMaRT to assess its effect over translation of target mRNAs, like in Figure 2.

Minor points:

1. The introduction begins stating that lncRNAs are expressed "in all cell types". I suggest adding a couple of words saying that many of them are shown to exhibit a very cell type specific expression pattern.
2. At the beginning of the Results section, when defining SMaRT, I suggest including the detail of how long is the mature lncRNA.
3. When mentioning the 3 bona fide protein interactors found by MS, DHX36, PURB and IQGAP1, I

suggest to define the 3 of them in that moment and at least to briefly mention in what the other two were found to participate.

4. In the discussion, I suggest to include some lessons from plant RNA biology, as it was shown in cotton that lincRNAs predominantly derive from LINE TEs (doi: 10.1186/s13059-018-1574-2).

5. Although I find that the MS is very well written, there are very few details that should be corrected. E.g., "This enzyme has been previously shown to bind" should be replaced by "This enzyme was/had been previously shown", in my opinion (I'm not a native speaker, anyway!).

1st Revision - authors' response

25 February 2020

Referee #1:

The Martone et al manuscript describes the long non coding RNA Inc-SMART that inhibits translation of G-quadruplex containing MLX-gamma mRNA by direct base-pairing with the G4-containing region of Mlx-gamma mRNA. This inhibition occurs by counteracting the effect of RNA helicase DHX36 that the authors identified as one of the Inc-SMART binding partners (likely indirect). As MLX proteins are myogenic transcription factors, which are tightly regulated during myogenesis. The authors show that the regulation of MLX protein translation and subcellular localization is dependent on Inc-SMART. Mis-expression of Inc-SMART by its knockdown or overexpression has a profound effect on normal differentiation of C2C12 cells.

While the interaction of IncSMART with both the G4-region of Mlx-gamma RNA and DHX36 RNA helicase was identified by the authors as a rather educated guess from their RNA pull-down experiments, the study is well-done and the main findings are well put together in one coherent story that merits publication in EMBO reports. My clarification requests and technical issues are listed below.

Major concerns:

1. My major request to the authors is that all measurements of transcript levels should be done by quantitative RT-PCR using at least three biological replicates. Data presented in the manuscript are often non-quantitative, lacking proper normalizations and controls and therefore are hard to interpret. In graph plots, it is not explained if biological or technical replicates are shown, which should be indicated in figure legends.

As requested, we changed semiquantitative RT-PCR with qPCR data. That was possible in Figure 3B and 3C, while in the case of Mlx (Figure 3D and 3E) this was not feasible since there are no oligos that allow the specific amplification of each individual isoform. The enrichment of Mlx was analyzed by qPCR only in psoralen cross-linking pull-down experiments presented in Figure EV3A where we wanted to test the overall population of Mlx transcripts. In the experiments where we wanted to distinguish between the three different isoforms (alfa, beta and gamma) we had to use semiquantitative PCR, as usually done for analyzing alternative splicing.

The gels are representative of single experiments that were repeated at least three times. We are sorry that by mistake this report was lacking; we have now added and highlighted all the informations about biological and technical replicates and statistical analysis in figure legends.

In addition, Figure 1E and 3A should indicate the number of independent biological samples and individual cells analysed.

We are sorry for the lack of such data. We have now added in the figure legends these informations. In particular, in Figure 1E the quantifications arise from at least 5 randomly chosen microscope fields of two independent biological samples. The samples were captured by inverted microscope Zeiss AxioObserver A1 equipped Plan-Neofluar EC 10×/0.3 M27 objective. Each microscope field contain about 600 cells.

In Figure 3A the image shows representative fields of immunofluorescence for total MLX protein. These localization analyses were performed on two independent biological samples, and about 50 individual cells were analyzed for each experimental condition (as now specified in the legend of Figure 3B)

2. Same for Fig2B, C (pull down and RIP) should show quantification with qRT-PCR rather than non-quantitative bands on the gel.

As requested, we changed semiquantitative RT-PCR with qPCR data (see new Figure 3B and 3C).

The specificity of the lncSMART-DHX36 interaction should be demonstrated by testing additional lncRNAs expressed at the levels similar to lnc-SMaRT.

The lncRNA Neat1 that, according to our RNA sequencing data, is expressed at comparable levels with lncSMaRT was selected and analyzed by qRT-PCR. This negative control has been added in Figure 3C (see also comment #13).

3. The authors should calculate and mention the approximate number of lnc-SMaRT molecules in C2C12 cells and *in vivo*. This information is key to understand the molecular mechanism of lnc-SMaRT especially because of its repressive effect on Mlx-gamma translation by direct base-pairing with its G4 region (which I guess happens stoichiometrically?) and sub-cellular localization of MLX proteins.

We thank the referee for pointing out the importance of stoichiometry in the interaction between RNA molecules. We indeed paid attention to this aspect and in fact it was one of the parameters used to pick up the Mlx mRNA as an interesting SMaRT interactor. The three different Mlx isoforms have altogether an expression level (RPKM 23.88) similar to that of lnc-SMaRT (RPKM 17.46), compatible with a likely interaction between these molecules.

As far as expression of SMaRT *in vivo*, we have to specify that it is not expressed in mature adult muscle fibers but only in regenerative conditions (see below response to comment n.6).

Finally, if genes are ranked based on decreased RPKM value in si-SCR conditions, lnc-SMaRT ranks 3027th out of 11643 transcripts, again indicating that its expression is medium-high. Since lnc-MD1, that we previously characterized (Legnini et al., 2015), is considered quite a fairly abundant species (RPKM of 69.9 and corresponding to approximately 600 copies/cell), we can conclude that lnc-SMaRT is a medium-high expressed lncRNA. We have added some numbers about this issue in the text as follows:

"Among them we selected the Mlx mRNA due to its known role in controlling myogenesis through the induction of several myokines. Moreover, such a transcript is present in three isoforms (α , β and γ) that have altogether an expression level (RPKM 23.88) similar to that of lnc-SMaRT (RPKM 17.46). All the three isoforms (α , β and γ) resulted enriched in the pull-down of lnc-SMaRT (Fig 3D)"

"This analysis showed that lnc-SmaRT (RPKM 17.46) is a medium-high abundant lncRNA being expressed only 3.6 times less than MyoD1 (RPKM 62.42) and 4 times less than lincMD1 (RPKM of 69.9) that is considered quite a fairly abundant muscle specific lncRNA (Legnini et al., 2015)."

In addition, it would be very informative to clarify the specificity of lncSMART pairing to the G4 region of Mlx-gamma mRNA i.e. would lncSMART inhibit translation of any RNA containing G4 elements in the 5'UTR (the authors mention that more than 100 G4-containing RNAs are known) or is the effect specific to Mlx-gamma only?

The RNAseq analyses on the RNAs recovered upon lncSMaRT pull-down allowed us to identify a list of 17 putative protein coding gene interactors and among them 12 resulted to encode for mRNAs with a predicted G-quadruplex overlapping the identified region of interaction with lncSMaRT (G-quadruplex prediction was performed using QGRS Mapper software; RNA-RNA interaction prediction was computed using IntaRNA 2.3.0). These data indicate that only a minor subset of G4 containing mRNAs stably interact with lncSMaRT. Moreover, in the case of Mlx we show a second region of interaction with SMaRT indicating that the interaction with the G4 region likely required some facilitator sequence or factor. This has been better specified in the discussion.

In conclusion, we believe that the translational regulation effect of lncSMaRT has a limited number of targets; further work will be necessary to apply the same type of approach used in this study to other mRNA targets.

Minor comments:

4. The authors should indicate the expression levels of the lnc-SMaRT relative to MyoD and myogenin.

As indicated in the previous comment (#3), lncSMaRT is fairly abundant RNA and its selection for further investigations was based also on this feature. From our characterization it reaches its peak of expression two days after switch to differentiation medium (RPKM 17.46) and in this condition, according to our RNA-seq data (TAB EV1), is only four times less expressed than MyoD1 (RPKM 62.42). This observation has been specified in the text together with the relative amount of lnc-SMaRT and Mlx mRNA (see comment #3).

5. Based on data presented in Fig 1 B, Fig EV1A, there is only one isoform lnc-SMaRT. The authors should mention if there are additional lncRNA isoforms or it is always only one mature lnc-SMaRT in C2C12; the size of the main mature lnc-SMaRT isoform should be mentioned in the text.

We added the length of the lncSMaRT main isoform in the manuscript.

As correctly suggested by the reviewer, two different isoforms of lncSMaRT, respectively 1409 nt and 653 nt long, have been annotated in the Ensembl database (see below). The difference between the two isoforms resides in the alternative first exon and in the length of the fourth exon (that in the short isoforms is 225 nt while in the long is 1080 nt), as indicated in the schematic representation below. We had previously performed the characterization of these isoforms to understand their relative amount and their expression profiles during C2C12 differentiation, from growth conditions to DM5. We performed a co-amplification of those species using a common reverse oligonucleotide (the base-pairing regions is shown by blue arrow) in combination with two forward oligonucleotides that are specific for each isoform (the base-pairing regions is shown by a purple arrow for the 1409 nt isoform and by an orange arrow for the 653 nt isoform). As shown in semiquantitative PCR the short isoform (upper band), is expressed only in DM2 and DM3 at very low levels while the long isoform reaches its peak of expression at day two, and it is still present at day 5. Due to the very low expression level observed for the short isoform we focused our study only on the longer one. Moreover, the described semiquantitative PCR showed the existence of a third unannotated isoform that differs from the longest one by the lack of the exon 3 (1272 nt long). Both isoforms (1409 nt and 1272 nt) have the same expression profile, are targeted by the siRNAs utilized in this study (indicated in the figure by red lines) and are amplified with the oligonucleotides that were used in qPCR as well as in semiquantitative PCR. Therefore, our functional studies include both isoforms. The location of PCR primers, that base pair with exon1 and exon2, has been added in the new figure 1.

Gene: Gm14635 ENSMUSG0000087591

Description predicted gene 14635 [Source:MGI Symbol;Acc:MGI:3705138]

Location Chromosome X: 12,339,782-12,354,996 reverse strand.
GRCm38:CM001013.2

About this gene This gene has 2 transcripts (splice variants) and 3 orthologues.

Transcripts Hide transcript table

Name	Transcript ID	bp	Protein	Biotype	CCDS	Flags
Gm14635-202	ENSMUST00000146535.7	1409	No protein	lncRNA	-	TSL:1 GENCODE basic
Gm14635-201	ENSMUST00000137281.1	643	No protein	lncRNA	-	TSL:2

6. "In vivo, lnc-SMaRT is present in conditions of muscle regeneration (Fig. EV1B), exemplified by skeletal muscles of dystrophic mdx mice". Does it mean that lncSMaRT is not expressed in the muscle under homeostatic conditions? This is a very interesting point and should be better explained in the main text.

As shown in Figure EV1B we were able to detect lncSMaRT expression *in vivo* only in skeletal muscles from *mdx* mice. As pointed out by the referee, we gave little emphasis to this observation. We rephrased the sentence in the following manner:

"The in vivo expression of lncSMaRT was analyzed in different tissues obtained from control and dystrophic mdx mice, which are characterized by high levels of muscle regeneration [23]. This condition was selected due to the observed in vitro involvement of lncSMaRT in early steps of myogenesis. PCR analyses showed that indeed the expression of lncSMaRT occurs only in mdx muscles while it is absent in skeletal and cardiac muscles of wild type mice, again suggesting lncSMaRT plays a role also in in vivo muscle regeneration (Fig EV1B)."

7. The authors should indicate the location of siRNAs they used for lncRNA-SMaRT KD in Figure 1A as well as the location of qPCR primers used in Figure 1D.

As suggested, we have added in Figure 1A the location of the siRNAs and qPCR primers used to deplete/amplify lncSMaRT. Primers are represented by blue arrows while the regions corresponding to the siRNAs are indicated by red lines. Specific sequences are available in the Methods section and in Table EV7.

8. Is downregulation of late differentiation markers shown in Fig EV1C significant? It should be indicated.

Yes, it is. Statistical analyses have been added to the new Figure (Figure EV1D) and to all those where such evaluations were missing.

9. GO analyses in Figure 2C should be presented in a form of a heatmap or a table with p-

values indicated for each term; from the current data presentation one cannot easily see ranking/p-values of all GO terms.

We added a supplementary table (Table EV2) containing detailed information about categories found in GO analysis. From the table, it can be fully appreciated that the categories that we highlight in the manuscript (muscle contraction and steroid biosynthesis for the down-regulated, cell proliferation for the up-regulated) are those with the highest rankings in terms of adjusted p-value.

10. Because the authors use siRNA to KD Inc-SMaRT which has both nuclear and cytoplasmic localization, the KD levels in the nucleus and cytoplasm should be shown as done in figure 1B

The Inc-SMaRT seems to work in the cytoplasm. This was shown not only by the translational effects observed but also by the luciferase experiments that imply a cytoplasmic activity. As suggested by the reviewer we performed such analysis. The results show that only the cytoplasmic form is affected, supporting the idea that IncSMaRT elicits its function in the cytoplasm. We present these data in the new Fig. EV1C.

11. I would suggest to change "a cDNA copy of Inc-SMaRT" in the first part of the results section to "mature Inc-SMaRT amplified from cDNA". What promoter was used for mature IncRNA overexpression?

According to Referee suggestion we modified the text in the following manner:

“Since plasmid transfection of C2C12 cells is very inefficient and could not provide suitable conditions for rescue experiments, we raised a stable C2C12 cell line overexpressing mature Inc-SMaRT amplified from cDNA under the control of eIF1a promoter.”

Further details about the cloning strategy are available in the Methods section.

12. In Fig EV2B, when showing sequence complementarity between Mlx and IncSMaRT transcripts, actual sequences should be shown (as supplementary figure), which are more informative than just a cartoon which is misleading as it shows complementarity of only a few nucleotides.

As requested, we added the sequences of complementarity between IncSMaRT and Mlx in the new Figure EV3C.

13. As already mentioned above, the anti-DHX36 RIP experiment in Fig 2E should be presented as a qRT-PCR and not as gel bands. Appropriate controls including DHX36 interaction with other lncRNAs should be included.

As explained in comment #1, this experiment cannot be presented as qRT-PCR because the alternative spliced isoforms of Mlx cannot be identified by this technique. Instead, following the reviewer' request we added appropriate controls in the new Figure 3C. The Neat1 lncRNA that, based on FPKM data, presented similar expression levels of lncSMaRT was used as a negative control together with the Rp7s mRNA, while the Wbp4 transcript, a known interactor of DHX36 (Lattmann et al., 2011), as a positive one. The enrichment of lncSMaRT is also shown.

14. The numbering of Fig EV2B and Fig EV2C is mixed up and should be fixed.

We thank the reviewer for pointing out this error that has been corrected in the new version of the manuscript following the new order of the Figures.

15. Western blot showing down-regulation of MLX-gamma upon lnc-SMaRT overexpression in HeLa cells should be quantified (Fig 2F). The number of independent replicates should be indicated.

As requested, the western blot bands shown in Figure 2F have been quantified and the corresponding histogram has been added to the panel (see the new Fig 3F). We also added the number of replicates (the experiment was performed at least three times and two replicates were quantified, as now stated in the figure legend).

16. In the left panel of Fig 4A, the constructs are not aligned with the results in the right panel, making the figure very confusing.

According to this suggestion we have adjusted panel A of the new Figure 5 as follow:

17. Fig 4B is not described in the text.

Probably this has escaped to the attention of the reviewer. The following paragraph was already present in the main text describing Figure 4B (now re-named 5B) and the related extended Figure EV4B (now re-named EV5B):

“To verify the contribution of DHX36 in the regulation of RLuc-Mlx 5’, the luciferase assay was performed in cells treated with siRNAs against DHX36, with or without the overexpression of lnc-SMaRT (Fig EV4B). As shown in Figure 4B, the luciferase expression was reduced when samples were depleted of DHX36, independently from the presence of lnc-SMaRT, without any change in RNA levels (Fig EV4B). These data indicate that DHX36 plays a positive role in the translational control of the G4 element. Interestingly, in conditions of DHX36 depletion, the presence of lnc-SMaRT was able to further decrease luciferase activity, indicating that base pairing per se plays a negative role on translation.”

Referee #2:

Martone and colleagues describe a lncRNA, named SMaRT, that is expressed in early phases of myoblast differentiation. Knockdown of SMaRT in C2C12 cells impairs myoblast differentiation. SMaRT is bound by DHX36, a protein involved in unwinding G4 structures. Among transcripts bound by SMaRT, the authors identify mRNAs with predicted G4 structures including MLX. Changes in SMaRT level affect MLX translation without impacting mRNA levels, in a DHX36 dependent manner. The authors propose SMaRT binds Mlx-g and impacts its translation and the nuclear localization of MLX proteins and transcriptional output. While the work is interesting further experiments to support the conclusions.

1- I have a general concern with data representation in the paper. Bar plots do not provide a full sense of the variability between the individual replicates and should be replaced by the actual individual values. A combination of bars and points would be suitable but as described elsewhere (Beyond Bar and Line Graphs: Time for a New Data Presentation Paradigm Tracey L. Weissgerber , Natasa M. Milic, Stacey J. Winham, Vesna D. Garovic Published: April 22, 2015 <https://doi.org/10.1371/journal.pbio.1002128>) replacing data points with bars representing SE or SD is not a faithful representation of the data. Related, instead of representative experiments, the data for the individual biological replicates should be provided as this gives a better sense of the actual variability between experiments. This might already be the case in some experiments but this is not specified and legends refer to "results of representative experiments". For western blots this could be achieved by quantifying the intensity. Uncropped blots don't seem to have been made available.

Data on individual experiments are now shown together with the bar plots for all the experiments. Uncropped blots have been added.

2- Do changes in Mef2C, Mck and Dys mRNA translate into changes in protein levels? Given the phenotype is unlikely to be driven by changes in mRNA this data is important.

We performed what requested and, as expected, the reduction in mRNAs expression is mirrored by changes in protein levels both for Mef2 and dystrophin: namely low levels of mRNAs are paralleled by low levels of proteins. We couldn't check for the Mck protein levels because of the unavailability

of the antibody; however, considering the results on the other two proteins we reasonably do not expect a different behavior for this factor.

3- Detailed information on the RNAseq data is missing. For example do gene expression differences between samples allow to distinguish siRNA and control treated samples (PCA)? What is the general extent of changes and significance (volcano plot)? What is the actual enrichment in different GO classes and their significance? The way the data is currently presented does not allow the reader to quantitatively assess the results.

MDS plot (similar to PCA, see new Fig. EV2A) and volcano plot have been added (see new Fig 2B). From the MDS plot, we can observe a separation between control cells and cells depleted of lnc-SMaRT, with the two siRNA#1-treated cell populations being more dissimilar.

As also requested by referee #1, we added a table describing the enrichment and significance of GO classes (Table EV2). From this table, it is evident that the categories that we highlight in the manuscript (muscle contraction and steroid biosynthesis for the down-regulated, cell proliferation for the up-regulated) are those with the highest rankings in terms of adjusted p-value.

4- To validate their siRNA results, the authors establish a C2C12 line overexpressing SMaRT (page3). The authors justify their experiments based on the low transfection efficiency of C2C12. While this is likely to be a problem, overexpression does not directly address the concern and cannot be used as validation. Notably while the levels of MyoD are not affected by KD they seem to be affected by overexpression. Generally, are the levels of affected genes anticorrelated between KD and OE cells? It's also unclear why proliferation and apoptosis is assessed in this cells but not in cells where SMaRT is KD. Given the siRNA transfection is efficient wouldn't OE of a siRNA resistant SMaRT be a better experiment.

We totally agree with the referee comment: the over-expression cannot be used as validation of the siRNAs experiment.

We performed the over-expression of a mutated form of SMaRT in which the sequence recognized by the siRNA#1 was deleted and tried to perform the classic rescue experiment. We also produced, as control, a construct for the over-expression of the full-length form of SMaRT. We noticed that the overexpression of SMaRT in growth conditions, where it is normally absent, produced an apoptotic phenotype, which made the rescue experiment impracticable. As the majority of lncRNAs, the expression of lncSMaRT is tightly regulated during differentiation and in the over-expressing cell line that we used, this regulation was lost. The constitutive promoter that we used, activated its expression in growth conditions where lncSMaRT is normally absent. For this reason, we get a different phenotype (and not opposite) from the one observed upon its depletion, such as in the case of MyoD expression. The apoptotic phenotype was not assessed in siRNA conditions because cell death was only observed upon the overexpression of SMaRT and we didn't performed RNA-seq analyses upon the overexpression of lncSMaRT (but an anticorrelation of affected genes upon KD is not expected).

We tried to improve the explanation of this point according to the referee's comment in the following manner:

In the attempt to perform a rescue experiment and since plasmid transfection of C2C12 cells is very inefficient, we raised a stable C2C12 cell line overexpressing mature lnc-SMaRT amplified from cDNA under the control of eIF1a promoter. Unfortunately, the overexpressing line displayed an apoptotic phenotype which hindered such type of analysis: in fact, myoblasts overexpressing the lncRNA were impeded to enter the myogenic program upon serum starvation by displaying only few elongated and oriented cells and decrease in mRNA and protein levels of MyoD and Myogenin (Fig

EV2D). Notably, at 48 hours of differentiation we observed an apoptotic phenotype, as indicated by increased activation of Caspase 3 (Fig 2D) and TUNEL assay (Fig EV2E). The increase in apoptosis and consequent reduction of the number of cells maintained in the myogenic program can explain the decrease in MyoD and Myogenin as well as the increase of transcripts related to cell proliferation observed upon lnc-SMaRT depletion. Altogether, these data indicate that the amount and timing of lnc-SMaRT expression should be finely regulated to establish a correct myogenic program.”

5- Related to the above point, if C2C12 transfect inefficiently how is this accounted for in the reporter experiments reported in Figure 4? Would this explain the relatively small effect sizes reported for the analysis reported in Fig 4?

Figure 4 corresponds to the new Figure 5.

In Figure 5A we used C2C12 cells to perform the experiment in a physiological context (muscle cells). Due to the low transfection efficiency we then moved to N2a cells (Figure 5B and C). It seems that the increase in the efficiency of transfection obtained in N2a cells is paralleled by a major decrease in RLuc-Mlx 5' luciferase activity upon the over expression of lncSMaRT as shown in Figure 4A #1 and #3 (27% of decrease) in comparison with Figure 5B (first two columns, 28% of decrease) and Figure 5C #1 and #2 (47% of decrease). Moreover, the decrease of the flagged version of Mlx γ observed upon the overexpression of lncSMaRT in HeLa cells, shown in Figure 3F, is similar (around 40%) to the one observed in N2a cells, suggesting that the decreased observed in C2C12 could be slightly underestimated.

6- Analysis of the RNA pull-down experiments should be more extensive (in line with comments for RNAseq) and made available.

Due to the peak-based target identification strategy, we could not draw PCA and volcano plot for the pull-down experiment. As extensively discussed in the answer to point 8 (see below), we performed a GO term enrichment analysis on the identified targets. Apart from the experimental validation of some of the targets, a proof of the success of the pull-down experiment is that we identified lnc-SMaRT as clearly enriched in pull-down with respect to both Input and LacZ.

7- The control RNA used in RNA- pull down (LacZ) has a very different GC content from that of endogenous mouse mRNAs. The impact on this on the pull-down results should be discussed.

We considered the possibility of a different GC content of LacZ while designing the different probe sets; therefore, we conceived the sets of oligos specific for lncSMaRT and for LacZ with similar GC composition (ranging from 40 to 55%) as reported in the following table.

We added this information by modifying the text with the following sentence:

*“To identify the binding partners of lnc-SMaRT, RNA pull-down experiments were performed with two sets of biotin-labeled DNA antisense oligonucleotides (Set#1 and Set#2, Fig 2A) on extracts derived from C2C12 cells at day 2 of differentiation. A set of antisense oligonucleotides against LacZ mRNA (LacZ), **with a similar GC content**, was used as a negative control.”*

Probe #	Probe (5' -> 3')	Probe Position	Percent GC
1	tagctagctccagtgactag	2	50.0%
2	aactagaacccccaaacaga	228	45.0%
3	cagcagttaggtccaattg	717	45.0%
4	tgggtaaagtgcttgatgca	905	45.0%

SMaRT probe set #1

SMaRT probe set #2	5	acacgggcatggtatacaac	1348	50.0%
	6	tgtgcctcatgtggggaag	87	55.0%
	7	taggaagcaagaccgtcatc	329	50.0%
	8	gtcttcgaggatcaaaggc	804	50.0%
	9	tactgctctcatcatttgc	985	40.0%
LacZ probe set	1	aatgtgagcgagtaacaacc		40.0%
	2	attaagtgggtaacgccag		45.0%
	3	aataattcgcgtctggcctt		45.0%
	4	aattcagacggcaaacgct		47.37%
	5	atcttcagataactgccgt		45.0%

8- The available analysis of the RNA recovered after SMaRT pull down is too superficial. Are mRNAs bound by the lncRNA part of a specific GO class? Do they have skewed GC contents? Length?

As suggested by the referee, we performed a GO term enrichment analysis on the top 30 genes and noticed that this list was enriched in transcripts encoding for ribosomal proteins, as can be seen in the following images.

Since these transcripts were also found in other pull-down experiments performed in our laboratory, we were afraid that they could likely represent false positives. This prompted us to re-evaluate the filter used in our target identification strategy, by directly comparing SMaRT pull-down reads against LacZ pull-down reads (previously the comparison with LacZ was only indirect, since we filtered out transcripts enriched both in lnc-SMaRT_VS_Input and LacZ_VS_Input comparisons). This procedure has now been described in methods section. The new list of targets, composed of 20 genes including SMaRT, 2 miRNAs and 17 protein-coding genes, is now described in the results and in the new Table EV4. As discussed in the results, we did not find any GO term enrichment in this list; furthermore, we significantly reduced the number of ribosomal proteins.

9- Of all the mRNAs that putatively interact with SMaRT, how many have a predicted G4? Is 18 more than expected by chance? Are the confidence scores of G4 predictions within this 18 mRNAs higher than for other mRNAs? Are this G4 also predicted by other methods?

Following the reviewer’s suggestion, we predicted G-Quadruplex in lncSMaRT pulldown mRNAs. We found that among the 17 mRNAs that were enriched in pulldown, 12 have a predicted G-Quadruplex. In order to analyze if the 70% of G-Quadruplex containing genes is more than expected by chance we compared pulldown enriched set to a control set of transcripts expressed in our system (C2C12 at day 2 of differentiation).

For every transcript enriched, derived from the same gene, we selected 10 control sets with the same number of transcripts from different genes. In order to avoid length bias between target and control sets, we checked that the number of genomic nucleotides covered was comparable (+/- 15 %).

After that, we predicted G-Quadruplex using QGRS software and we compared the number of G-Quadruplex containing genes with Fisher’s exact test. Although G-Quadruplex containing genes in the pulldown set are 70% and in the control sets are 58% we were not able to see a statistical significant enrichment (pvalue = 0.44). In order to analyze the propensity of pulldown and control sets to form G-Quadruplex, we also compared the G4 confidence score of predictions with Mann-Whitney U test. Also in this case, although the G-Score was higher in pulldown set (mean= 49 , median= 62) compared to control sets (mean = 45, median = 57), we were not able to see a statistical significance (pvalue = 0.057).

See below for boxplot representing G-score distributions of pulldown set and control sets:

As suggested by the referee, we performed G-Quadruplex prediction on lnc-SMaRT interactors with other available methods. Briefly, the existing software for computation G-Quadruplex identification are based on different algorithms: Regular expression matching, Scoring, Sliding windows and Machine learning algorithms (Puig Lombardi, E., & Londoño-Vallejo, A. (2020). A guide to computational methods for G-quadruplex prediction. *Nucleic Acids Research*, 48(1), 1–15. <https://doi.org/10.1093/nar/gkz1097>). QGRS mapper, the tool that we used in the work, is based on regular expression matching and scoring of predicted G-quadruplexes. To confirm the prediction with the other algorithms, we selected G4 Hunter” (Bedrat, A., Lacroix, L., & Mergny, J. L. (2016). Re-evaluation of G-quadruplex propensity with G4Hunter. *Nucleic Acids Research*, 44(4), 1746–1759. <https://doi.org/10.1093/nar/gkw006>), a Sliding window-Scoring program, and “G4 RNA screener” (Garant, J. M., Perreault, J. P., & Scott, M. S. (2017). Motif independent identification of potential RNA G-quadruplexes by G4RNA screener. *Bioinformatics*, 33(22), 3532–3537. <https://doi.org/10.1093/bioinformatics/btx498>) a Machine learning algorithm.

The following table reports the comparison of G-quadruplexes identified by QGRS mapper and by the other two tool with default parameters. Of note, 9 G-quadruplexes from 6 genes are confirmed by both the other softwares, while other 13 G-quadruplexes from 7 genes are confirmed by at least one of the used tools.

Gene	ID	start	end	G4	Gs	Localization	G4Hunter	G4RNA Screener
------	----	-------	-----	----	----	--------------	----------	----------------

Acad8	ENSMUST00000060513	299	334	GGGATTGGGGGGTCTATGTGCGAACAGATGTGGG	52	CDS	YES	YES
Acad8	ENSMUST00000120367	316	351	GGGATTGGGGGGTCTATGTGCGAACAGATGTGGG	52	CDS	NO	YES
Acs16	ENSMUST00000108905	24	48	GGGGCTCGGGGGCTCGGGCCCTGGG	62	5UTR	YES	YES
Acs16	ENSMUST00000127731	2807	2838	GGGTTGGGATTCTGGGTGTTCTCCATGGAGGG	56	/	NO	NO
Acs16	ENSMUST00000127731	2615	2654	GGGTGGGATGGGGTAGTTCATGTCTAGGGTTGAGAGTGGG	60	/	YES	YES
Acs16	ENSMUST00000108904	24	48	GGGGCTCGGGGGCTCGGGCCCTGGG	62	5UTR	YES	YES
Arf5	ENSMUST00000020717	1013	1056	GGGGGTACCTTGGGGCCAGTTTTGGGGGGAGGAAAGTGAGGG	63	3UTR	YES	YES
Coq2	ENSMUST00000126981	20	51	GGGAGGCGGGGGCTCGCGGGGGCTCGCGG	63	/	YES	YES
Coq2	ENSMUST00000126981	114	155	GGGGTTCCGGGCGCGGGGATCGGCGAGCCCCGCCCCCGGG	54	/	NO	NO
Coq2	ENSMUST00000135146	21	62	GGGGTTCCGGGCGCGGGGATCGGCGAGCCCCGCCCCCGGG	54	/	NO	NO
Coq2	ENSMUST00000031262	55	86	GGGAGGCGGGGGCTCGCGGGGGCTCGCGG	63	CDS	YES	YES
Coq2	ENSMUST00000031262	149	190	GGGGTTCCGGGCGCGGGGATCGGCGAGCCCCGCCCCCGGG	54	CDS	NO	NO
Glis3	ENSMUST00000065113	2048	2086	GGGCACTCCCCAGGGCCGGGCCTGGGCCAGGGCCTGGG	64	/	NO	YES
Glis3	ENSMUST00000162022	2667	2705	GGGCACTCCCCAGGGCCGGGCCTGGGCCAGGGCCTGGG	64	CDS	NO	YES
Glis3	ENSMUST00000162022	7280	7322	GGGGATGGTGATTATAATTAAGCAGATGGGGGGGAAGGGG	67	3UTR	YES	YES
Glis3	ENSMUST00000161026	1713	1751	GGGCACTCCCCAGGGCCGGGCCTGGGCCAGGGCCTGGG	64	/	NO	YES
Mlx	ENSMUST00000017945	98	114	GGGGAGGGCGGGTTCGGG	63	CDS	YES	YES
Myo1c	ENSMUST00000102505	4451	4485	GGGTGCCTCTGTGACCTGGGAGCCTAGGGACAGGG	56	3UTR	NO	YES
Myo1c	ENSMUST00000108431	4519	4553	GGGTGCCTCTGTGACCTGGGAGCCTAGGGACAGGG	56	3UTR	NO	YES
Myo1c	ENSMUST00000108431	4402	4422	GGGAGCTACCGGGTGGGAGGG	60	3UTR	NO	NO
Myo1c	ENSMUST00000108431	280	324	GGGCAGCGGAGCGGGCGCCGGTCCGGCAGGATGCGCTACCGGG	54	5UTR-CDS	NO	YES
Myo1c	ENSMUST00000108431	200	230	GGGGCCTGCAAGGGCGGTGCAGGGGGCGGG	60	5UTR	NO	NO
Myo1c	ENSMUST00000102504	4458	4492	GGGTGCCTCTGTGACCTGGGAGCCTAGGGACAGGG	56	3UTR	NO	YES
Myo1c	ENSMUST00000069057	4229	4249	GGGAGCTACCGGGTGGGAGGG	60	3UTR	NO	NO
Myo1c	ENSMUST00000069057	4346	4380	GGGTGCCTCTGTGACCTGGGAGCCTAGGGACAGGG	56	3UTR	NO	YES
Myo1c	ENSMUST00000102505	4334	4354	GGGAGCTACCGGGTGGGAGGG	60	3UTR	NO	NO
Myo1c	ENSMUST00000102504	4341	4361	GGGAGCTACCGGGTGGGAGGG	60	3UTR	NO	NO
Ndr4	ENSMUST00000041318	2377	2413	GGGCTGGAGATTGCCTGGCCCTGGGTGGGAAATGGG	51	3UTR	NO	NO
Ndr4	ENSMUST00000080666	2014	2050	GGGCTGGAGATTGCCTGGCCCTGGGTGGGAAATGGG	51	3UTR	NO	NO
Ndr4	ENSMUST00000166358	2243	2279	GGGCTGGAGATTGCCTGGCCCTGGGTGGGAAATGGG	51	/	NO	NO
Ndr4	ENSMUST00000073139	2079	2115	GGGCTGGAGATTGCCTGGCCCTGGGTGGGAAATGGG	51	3UTR	NO	NO
Rbm45	ENSMUST00000046389	41	75	GGGGCGAGACGGGAGCTCCGGGAAGCGGCCGGG	63	5UTR	NO	YES
Six1	ENSMUST00000050029	329	366	GGGGCGGAGGGTGGCGCGCTTTGCTCCGGGCCGGG	53	5UTR	NO	NO
Six1	ENSMUST00000050029	1599	1635	GGGTTCTAAGTGGGGAGATATTGGGCCTTGAAGGG	63	CDS-3UTR	NO	YES
Spire1	ENSMUST00000115050	253	297	GGGCCCCGTTCTGGGTACAAGTATGAGGGATTTCGAAATGGGG	62	CDS	NO	NO
Spire1	ENSMUST00000082243	407	451	GGGCCCCGTTCTGGGTACAAGTATGAGGGATTTCGAAATGGGG	62	CDS	NO	NO
Spire1	ENSMUST00000045105	351	395	GGGCCCCGTTCTGGGTACAAGTATGAGGGATTTCGAAATGGGG	62	CDS	NO	NO
Usp10	ENSMUST00000144458	2260	2295	GGGCAAGGGCAGCGAGGACGAGTGGGAGCAAGTGGG	56	CDS	NO	YES
Usp10	ENSMUST00000108988	1803	1838	GGGCAAGGGCAGCGAGGACGAGTGGGAGCAAGTGGG	56	CDS	NO	YES

10- Is there a G4 sequence in SMaRT

As specified in the main text there are no trustable G-quadruplex forming sequences on lncSMaRT. lncSMaRT sequence was analyzed on composition and distribution of putative Quadruplex forming G-Rich Sequences (QGRS) with two different quadruplex-predicting software: "QGRS Mapper"

(Kikin et al., 2006) and “QuadBase2” (Dhapola et al., 2016); two regions prone to fold in this manner were found according to QGRS Mapper, but with a very low G-score (10 and 18 respectively) and their presence was not confirmed by “QuadBase2” tool.

This was already specified in the text:

“The search for putative G-quadruplex forming sequences in Inc-SMaRT with the QGRS Mapper software [26] and with the “QuadBase2” tool [41] did not predict the occurrence of bona fide G4-elements. Therefore, it is likely that the interaction of Inc-SMaRT and DHX36 is indirect”.

11- What happens to Mlx-g translation and MLX localization in C2C12 SMaRT overexpressing cells?

In order to test the effect of Inc-SMaRT overexpression on the subcellular localization of endogenous MLX proteins, we performed immunolocalization experiments analogous to the ones presented in the main Figure 4A (C2C12 cells at day 2 of differentiation) but using the stable C2C12 cell line overexpressing IncSMaRT. As already observed in Figure EV2-E and discussed in the text, Inc-SMaRT overexpression has a toxic effect producing a clear apoptotic response and a strong inhibition of myoblast differentiation, with very few cells able to enter the myogenic program. In these conditions, a significant analysis concerning Mlx subcellular localization and translation was not possible, since only few cells were MHC positive (see Figure below).

12- Are MLX target genes enriched among genes differentially expressed upon SMaRT knockdown?

We crossed the list of genes deregulated after Inc-SMaRT depletion and the list of MLX targets we found in a work by Hunt et al. 2015 (<https://www.ncbi.nlm.nih.gov/pmc/articles/PMC4691951/>). Since SMaRT represses Mlx translation, we were interested in deregulated genes having opposite behavior upon the knock-down of Inc-SMaRT and MLX. We identified 10 of them, which statistically support a significant overlap between the two lists and serves as evidence that the MLX translational regulation mediated by SMaRT results in the deregulation of MLX targets. This result has been added in the new Fig 4C.

The text has been modified as follow:

“Finally, the link between nuclear localization of MLX and its transcriptional activity was tested by intersecting its target genes in C2C12 cells with those deregulated upon Inc-SMaRT depletion. We found that the expression of a significant number of genes changed in the opposite direction upon the depletion of MLX and Inc-SMaRT (Fig 4C). qPCR analysis of CCL2 and CCL7, which are up-

regulated by Mlx, showed their up-regulation upon lnc-SMaRT depletion at day 2 of differentiation (Fig EV4F and Table EV1). In conclusion, these results suggest that lnc-SMaRT could act as a repressor of Mlx- γ , and that in turn this directly affects the nuclear localization of total MLX proteins and their transcriptional output.

13- Addition of DHX36 should impact interaction between G4-Mlx-g and complementary region in SMaRT (Figure 4).

Figure 4 corresponds to the new Figure 6.

We tried to set up a protocol for the over-expression and purification of DHX36 using FLAG tag, but we didn't succeed. To answer this question, circumventing the problem, we performed the experiment shown in Figure 6D and E in which the high temperature treatment followed by slow renaturation should mimics the action of DHX36 in solving the G4 structure present on the γ -oligo, favoring its interaction with the SMaRT oligo. In Figure 6E lane 4, in the absence of the "heat pulse", the G-quadruplex structure formed by the γ -oligo is well detectable (indicated by the arrow γ -G4), while it is decreased in lane 5 where the interaction among the two oligos is favored by the "heat pulse" to the detriment of the formation of the quadruplex. Moreover, an intermediate situation is observed when the G-quadruplex formation is stabilized by KCl supplementation in the reaction.

Other comments

14- Since SMaRT appears to function in a G4 dependent manner, the topic is not covered in the introduction. It is mentioned in the discussion but to understand the results the reader needs to know what G4 are, the proteins involved in their resolution and examples of other G4-related lncRNAs.

Since the involvement of lnc-SMaRT in G4 regulation is the finding of the work, we thought to include these notes in the discussion where an extensive description is reported. Following the reviewer's comment, we have introduced a sentence in the introduction:

"An important component in translational regulation is represented by G-quadruplex regions, which are non-canonical secondary structures that form within G-rich DNA or RNA sequences by Hoogsteen hydrogen bonds (Fay et al., 2017) known to require specific helicases to be resolved (Sauer et al., 2017)."

Minor Comments

15- It would be useful to mention early on that SMaRT was identified in mouse.

We anticipated this information adding the word “murine” in the abstract as well as the first section of results as indicated below:

Abstract

*Here we describe a regulatory circuitry operating in the early phases of **murine** muscle differentiation in which a long non coding RNA (SMaRT) base pairs with a G4-containing mRNA (Mlx- γ) and represses its translation by counteracting the RNA helicase DHX36 action....*

Lnc-SMaRT depletion affects myoblast differentiation

*Lnc-SMaRT (Skeletal Muscle Regulator of Translation) is an intergenic long noncoding RNA, previously named lnc-049 (MGI Symbol: GM14635), identified as a **murine** skeletal muscle species [19].*

Referee #3:

The work by Martone and co-workers shed light on the role of the regulatory lncRNA SMaRT, which is able to form G4-mediated RNA duplexes with target mRNAs. The MS is generally very well written and it is easy to follow albeit the high methodology complexity and the model proposed. Shortly, the SMaRT noncoding transcript was first identified in the context of myogenesis. The authors found that SMaRT can directly interact with target mRNAs forming duplexes and modulating translation. This finding is based on a SMaRT-pull down assay using independent interspersed sets of biotinylated probes, and compared to an anti-LacZ set of primers used as a negative control. SMaRT-pulldown was followed by sequencing for associated RNAs identification. In particular, SMaRT can recognize in vivo three alternative isoforms of the Mlx mRNA. However, SMaRT can only impair the translation of the Mlx- γ isoform, with which it can interact with an additional region present in the mRNA forming a G4 complex. Furthermore, the authors performed a SMaRT-pulldown followed by protein precipitation and MS. This approach allowed the identification of DHX36 as a potential interactor, an enzyme capable of recognizing and unwinding G-quadruplex structures. The protein-RNA interaction was accurately confirmed by alternative methods, e.g. RIP using specific antibodies. Although G4 RNA structures were previously associated to several molecular mechanisms including translational modulation, in this work Martone and co-authors show a very compelling case of an inter-RNA G4 structure mediating the lncRNA regulation over its target mRNA counterpart. Moreover, the authors shed light on the impact of SMaRT-MLX- γ interaction over translation of specific mRNA isoforms and the sub-cellular distribution of the translated Mlx pool of proteins. In this part, I should say that while reading the MS I had doubts about the conclusions drawn by the authors based on the FLAG fusions. I first thought that if the FLAG tag was fused to the end of the gene, then truncated FLAG-free proteins may be produced. However, reading carefully the M&M section I found out that the FLAG was fused to the 5' end of the gene, discarding this possibility. I suggest the authors to include a phrase in the figure legend or scheme in the figure itself explaining the fusion construct.

This has been clarified in the legend of Figure 3F.

I'd like to suggest a few more points to improve the MS:

Major point:

Although the rationale of the MS is easy to follow and the order of the experiments is logical, I find that the first results could be strengthened based on the later ones. In particular, Figure 2F shows the effect of SMaRT over Mlx translation. Later in the MS the authors characterize what part of the lncRNA matches its target mRNA using a very elegant reporter-based approach and in vitro assays. I wonder why the authors didn't try to over express the different mutant versions of SmaRT to assess its effect over translation of target mRNAs, like in Figure 2.

The lnc-SmaRT mutant available is the one where we swapped the region of complementarity (region A) with the G4 element of MLX (see Fig 5C). Unfortunately, this is not an appropriate control to use in the experiment of Figure 2F since the high expression levels obtained upon transfection could sponge DHX36 and anyhow interfere with Mlx translation. In the luciferase experiment of Fig 5C this construct was tested in combination with the complementary mutation in Mlx aiming at demonstrating the relevance of pairing for the translational repression. Instead in the experiment of Fig 2F the Mlx counterpart could not pair with this lnc-SmaRT mutant; therefore, becoming sensible to the competition for the DHX36 binding and anyhow being repressed translationally.

Minor points:

1. The introduction begins stating that lncRNAs are expressed "in all cell types". I suggest adding a couple of words saying that many of them are shown to exhibit a very cell type specific expression pattern.

The reviewer is correct and following his/her suggestion we have modified the beginning of the introduction as follow:

“Long non coding RNAs (lncRNAs) belong to a complex class of transcripts which are expressed in all cell types and can be considered as key regulators of development and differentiation, thanks to the exquisite regulation of their spatiotemporal pattern of expression (Batista and Chang, 2013; Fatica and Bozzoni, 2014). Specific functions have been defined only for some of them so far, despite the fact that they represent a large fraction of the mammalian transcriptome. Loss and gain of function experiments indicated that they may play an important role in normal development and differentiation as well as in many pathological conditions [1–3];...”

2. At the beginning of the Results section, when defining SMaRT, I suggest including the detail of how long is the mature lncRNA.

We have added the length of lncSMaRT main isoform in the manuscript.

3. When mentioning the 3 bona fide protein interactors found by MS, DHX36, PURB and IQGA1, I suggest to define the 3 of them in that moment and at least to briefly mention in what the other two were found to participate.

Following the reviewer's suggestion, we have added the description of the other two protein candidates. Please check the modified sentence:

“The MS data allowed to short list three bona fide interactors (PURB, IQGA1 and DHX36) based on the number of unique peptides (more than 5) and on the enrichment with both sets of specific oligos in comparison to LacZ probes. PUR β is a single-stranded DNA- and RNA-binding protein that has been previously involved in DNA replication/transcription and in mRNA translational efficiency attenuation (Gupta et al., 2003) while IQGA1 is a Ras GTPase-activating-like protein that belongs to a family of scaffolding proteins involved in several cellular processes such as cell cycle regulation, cell-cell adhesion and actin cytoskeleton organization (Brown et al., 2006). DHX36 is an ATP-

dependent RNA helicase and was selected for further investigations because of the absence of peptides in the LacZ sample (Table EV2). ..."

4. In the discussion, I suggest to include some lessons from plant RNA biology, as it was shown in cotton that lincRNAs predominantly derive from LINE TEs (doi: 10.1186/s13059-018-1574-2).

We thank the reviewer for this useful suggestion; we have followed this advice and added the quotation.

5. Although I find that the MS is very well written, there are very few details that should be corrected. E.g., "This enzyme has been previously shown to bind" should be replaced by "This enzyme was/had been previously shown", in my opinion (I'm not a native speaker, anyway!).

Thank you for the suggestion.

Thank you for the submission of your revised manuscript. We have now received the enclosed reports from the referees that were asked to assess it. Referee 2 still has a minor suggestion that you can incorporate (if you agree) before we will proceed with the official acceptance of your manuscript.

A few minor other changes are also required.

Tables EV1, 2 and 3 should rather be called Dataset EV1, EV2, EV3. Please also avoid splitting a table into A and B. You could either combine the 2 files in an excel file with more tabs, or give the 2 tables 2 different numbers. Please also correct all callouts in the manuscript file for the EV tables that are changed to Datasets.

Table EV4, 5, 6 and 7 can remain EV tables, but please combine EV4A and EV4B into either EV4 or split into EV4 and EV5. Please also make sure here that all callouts in the manuscript are corrected.

I would like to suggest a few minor changes to the abstract. Do you agree with :

Guanine-quadruplexes (G4) included in RNA molecules exert several functions in controlling gene expression at post-transcriptional level; however, the molecular mechanisms of G4-mediated regulation are still poorly understood. Here we describe a regulatory circuitry operating in the early phases of murine muscle differentiation in which a long non coding RNA (SMaRT) base pairs with a G4-containing mRNA (Mlx-gamma) and represses its translation by counteracting the activity of the DHX36 RNA helicase. The time-restricted, specific effect of lnc-SMaRT on the translation of the Mlx-gamma isoform modulates the general subcellular localization of total MLX proteins, impacting on their transcriptional output and promoting proper myogenesis and mature myotube formation. Therefore, the circuitry made of lnc-SMaRT, Mlx-gamma and DHX36 not only plays an important role in the control of myogenesis but also unravels a molecular mechanism where G4 structures and G4 unwinding activities are regulated in living cells.

EMBO press papers are accompanied online by A) a short (1-2 sentences) summary of the findings and their significance, B) 2-3 bullet points highlighting key results and C) a synopsis image that is 550x200-400 pixels large (the height is variable). You can either show a model or key data in the synopsis image. Please note that text needs to be readable at the final size. Please send us this information along with the revised manuscript.

Best wishes and stay safe,

REFeree REPORTS

Referee #1:

The authors adequately addressed my previous concerns and questions and significantly improved their data presentation.

Referee #2:

I appreciate the effort the authors have putted into addressing my concerns and clarifying the points raised by me and other reviewers regarding data presentation and addition of supporting data. I also appreciate the extra analysis that was done in response to my questions. In my opinion, the results of this analysis should be added to the manuscript, even when it provides results that are not fully supportive of the authors model (for example analysis of point 9).

2nd Revision - authors' response

26 March 2020

The authors performed all minor editorial changes.

Corresponding Author Name: Irene Bozzoni

Manuscript Number: EMBOR-2019-49942V1